# EduLLM: Leveraging Large Language Models and Framelet-Based Signed Hypergraph Neural Networks for Student Performance Prediction

Ming Li [1]   Yukang Cheng [1]   Lu Bai [2]   Feilong Cao [3]   Ke Lu [4,5]   Jiye Liang [6]   Pietro Lio [7]

## Abstract

The growing demand for personalized learning underscores the importance of accurately predicting students' future performance to support tailored education and optimize instructional strategies. Traditional approaches predominantly focus on temporal modeling using historical response records and learning trajectories. While effective, these methods often fall short in capturing the intricate interactions between students and learning content, as well as the subtle semantics of these interactions. To address these gaps, we present EduLLM, the first framework to leverage large language models in combination with hypergraph learning for student performance prediction. The framework incorporates FraS-HNN (Framelet-based Signed Hypergraph Neural Networks), a novel spectral-based model for signed hypergraph learning, designed to model interactions between students and multiple-choice questions. In this setup, students and questions are represented as nodes, while response records are encoded as positive and negative signed hyperedges, effectively capturing both structural and semantic intricacies of personalized learning behaviors. FraS-HNN employs framelet-based low-pass and high-pass filters to extract multi-frequency features. EduLLM integrates fine-grained semantic features derived from LLMs, synergizing with signed hypergraph representations to enhance prediction accuracy. Extensive experiments conducted on multiple educational datasets demonstrate that EduLLM significantly outperforms state-of-the-art baselines, validating the novel integration of LLMs with FraS-HNN for signed hypergraph learning.

## 1. Introduction

**Background.** In recent years, the increasing emphasis on personalized learning has underscored the need for innovative strategies to support individualized educational pathways (Mayer, 2019). A critical aspect of this effort is the accurate prediction of student performance, which enables educators to tailor instructional strategies, optimize learning experiences, and improve educational outcomes (Dziuban et al., 2015). While adaptive learning technologies have demonstrated promise, they often struggle to capture the intricate structural and semantic relationships inherent in the learning process, such as interactions between students and educational content.

**Motivation.** Existing approaches to student performance prediction predominantly rely on temporal modeling and graph neural networks (GNNs) to analyze historical answering records, learning trajectories, and pairwise connections between learners and learning materials. Temporal models focus on sequential dependencies, while GNNs excel at representing "student-content" interactions (Zhou et al., 2020; Zhang & Chen, 2018). These methods have achieved considerable success by capturing local relationships within the learning ecosystem. However, their reliance on pairwise modeling limits their capacity to represent higher-order interactions involving multiple entities, such as a student's engagement with related questions or the interconnectedness of various knowledge points within a course.

Hypergraph learning offers a compelling alternative by extending traditional graph structures to model higher-order relationships (Antelmi et al., 2023; Dai & Gao, 2023; Kim et al., 2024). Unlike GNNs, hypergraphs can represent in-

---

[1]Zhejiang Key Laboratory of Intelligent Education Technology and Application Zhejiang Normal University, Jinhua, China [2]School of Artifcial Intelligence, and Engineering Research Center of Intelligent Technology and Educational Application, Ministry of Education, Beijing Normal University, Beijing, China [3]School of Mathematical Sciences, Zhejiang Normal University, Jinhua, China [4]School of Engineering Science, University of Chinese Academy of Sciences, Beijing, China [5]Peng Cheng Laboratory, Shenzhen, China [6]Key Laboratory of Computational Intelligence and Chinese Information Processing of Ministry of Education, the School of Computer and Information Technology, Shanxi University, Taiyuan, China [7]Department of Computer Science and Technology, University of Cambridge, Cambridge, UK. Correspondence to: Lu Bai <bailu@bnu.edu.cn>.

*Proceedings of the 42nd International Conference on Machine Learning*, Vancouver, Canada. PMLR 267, 2025. Copyright 2025 by the author(s).

teractions among multiple nodes simultaneously through hyperedges, providing a richer representation of complex relationships (Li et al., 2024). For instance, a student's responses to multiple-choice questions (MCQs) can be captured as hyperedges, effectively modeling both structural and semantic associations. Despite this potential, the application of hypergraph-based methods in student performance prediction remains limited, particularly in contexts that require the integration of semantic insights and structural information.

Meanwhile, the advent of large language models (LLMs) has marked a transformative shift in natural language processing. LLMs excel at extracting fine-grained semantic features from textual data, such as the content of MCQs, and offer a complementary approach to hypergraph learning by enhancing the semantic granularity of question representations (Wang et al., 2024b). However, existing methods struggle to effectively integrate LLMs with hypergraph-based approaches, leading to feature embeddings that are noisy or fail to capture the full complexity of student-content interactions. Furthermore, many hypergraph learning frameworks lack mechanisms for multi-scale representation, which are critical for capturing both shared preferences among students and individualized distinctions in learning behaviors.

**Our Methodology.** To address these challenges, this paper introduces EduLLM, a unified framework that integrates hypergraph learning with LLMs to enhance the precision of student performance prediction. At its core, EduLLM formulates the student performance prediction task as a hypergraph learning problem, where interactions between students and MCQs are modeled using signed hypergraphs. In this formulation, students and questions are represented as nodes, and their response records are encoded as positive and negative signed hyperedges, capturing both structural and semantic intricacies of the learning process. A key innovation of EduLLM lies in its framelet-based signed hypergraph neural network (FraS-HNN), which leverages multiresolution signal processing to effectively learn hypergraph representations. This model employs low-pass filters to identify shared patterns in learning behaviors and high-pass filters to highlight individual differences, providing a nuanced understanding of student-content interactions. Framelet-based transformations further enable decomposition and reconstruction in the frequency domain, enhancing the model's ability to capture multi-scale features while reducing noise and improving data compression. Fine-grained semantic features extracted by LLMs are seamlessly integrated into this framework, enriching the representation of course content and improving prediction accuracy. Extensive experiments on multiple educational datasets demonstrate that EduLLM outperforms state-of-the-art baselines, validating the effectiveness of combining hypergraph learning and LLM-based semantic analysis. By addressing limita-

tions in existing approaches, this framework provides robust technological support for advancing personalized learning and facilitating precise educational interventions.

**Contribution**. The primary contributions of this work are summarized as follows:

- **Problem Formulation Perspective**: We redefine the student performance prediction task as a hypergraph learning problem by introducing signed hypergraphs to represent interactions between students and questions, where response records are modeled as positive and negative signed hyperedges.

- **Model Development**: We design a framelet-based signed hypergraph neural network (FraS-HNN) that leverages multiresolution signal processing to effectively learn hypergraph representations. This model employs low-pass filters to capture shared preferences among learners and high-pass filters to emphasize individual distinctions, facilitating a nuanced understanding of both structural and semantic relationships within the signed hypergraph.

- **Framework Design**: We propose EduLLM, a unified framework that synergizes hypergraph learning and LLM-based semantic analysis. By combining hypergraph-based structural modeling with the semantic granularity provided by LLMs, EduLLM achieves a comprehensive representation of student-content interactions. This integration significantly improves the precision of student performance predictions, demonstrating the effectiveness of the framework across multiple datasets.

## 2. Problem Formulation and Notation

In this section, we introduce how the interactions between students and multiple-choice questions (MCQs) can be represented using a signed hypergraph. The interaction data includes not only students' responses to MCQs but also detailed textual information associated with each question, such as the question stem, options, answers, and explanations. These interactions are naturally modeled as a signed hypergraph, where nodes represent students and questions, and hyperedges capture the relationships between students and specific questions based on their responses.

**Signed Hypergraph.** The signed hypergraph is mathematically defined as $\mathcal{G} = (\mathcal{V}, \mathcal{E})$, where $\mathcal{V} = \mathcal{U} \cup \mathcal{Q}$ represents the set of nodes, comprising the student set $\mathcal{U}$ and the question set $\mathcal{Q}$. The hyperedge set $\mathcal{E}$ consists of two components: the positive hyperedge set $\mathcal{E}^+$ and the negative hyperedge set $\mathcal{E}^-$. Specifically, a positive hyperedge $e^+ \in \mathcal{E}^+$ connects all students who answered a question correctly, while

a negative hyperedge $e^- \in \mathcal{E}^-$ connects all students who answered the same question incorrectly.

To formally describe the structure of the signed hypergraph, we use an incidence matrix $H \in \{0, 1, -1\}^{N \times M}$ where $N = |\mathcal{V}|$ is the number of nodes and $M = |\mathcal{E}|$ is the number of hyperedges. The matrix element $H(v, e)$ encodes the relationship between a node $v \in \mathcal{V}$ and a hyperedge $e \in \mathcal{E}$: (i) $H(v, e^+) = 1$, if $v$ belongs to a positive hyperedge $e^+$; (ii) $H(v, e^-) = -1$, if $v$ belongs to a negative hyperedge $e^-$; (iii) $H(v, e) = 0$ otherwise.

The degrees of nodes and hyperedges are represented by the diagonal matrices $D_v \in \mathbb{R}^{N \times N}$ and $D_e \in \mathbb{R}^{M \times M}$, respectively, as computed as follows:

$$D_v(v,v) = \sum_{e \in \mathcal{E}} |H(v,e)|, \; D_e(e,e) = \sum_{v \in \mathcal{V}} |H(v,e)|. \quad (1)$$

**Task Definition.** Within this signed hypergraph framework, the student performance prediction task is formulated as a hyperedge sign prediction problem. Here, the objective is to predict the unknown signs of hyperedges, representing whether students correctly or incorrectly respond to specific questions. Given a signed hypergraph $\mathcal{G} = (\mathcal{V}, \mathcal{E})$, we aim to learn low-dimensional embeddings for students and questions, denoted as $\mathbf{z}_{u_i}$ for student $u_i \in \mathcal{U}$ and $\mathbf{w}_{q_j}$ for question $q_j$, where $\mathbf{z}_{u_i}, \mathbf{w}_{q_j} \in \mathbb{R}^d$ and $d$ is the embedding dimension. The embeddings are used to predict the sign of an unknown hyperedge $e_{i,j}$ via a mapping function: $f(\mathbf{z}_{u_i}, \mathbf{w}_{q_j}) \rightarrow -1/+1$.

**Remark 1.** We note that our problem formulation, i.e., modeling student–question interactions through signed hypergraphs, differs fundamentally from conventional knowledge tracing (KT) (Abdelrahman et al., 2023) and cognitive diagnosis (CD) (Wang et al., 2024a) frameworks. KT and CD are typically centered on tracking students' mastery of underlying concepts over time and rely heavily on fine-grained mappings between questions and predefined concepts. In contrast, our formulation operates at the question level, leveraging signed hyperedges to encode the correctness of student responses. This representation is inherently incompatible with the structure and assumptions of most KT/CD datasets, which are not designed to support signed or higher-order relational modeling at the question level. Technically, how to formulate signed hypergraphs within the KT/CD setting, particularly in the context of modeling concept-level mastery via Q-matrices, requires further investigation. We leave this as a potential direction for future work to connect signed hypergraph learning with traditional student modeling paradigms.

**Remark 2.** The signed hypergraph framework offers a significant advantage over traditional signed graphs by its ability to model higher-order relationships among students and questions. While signed graphs are limited to pairwise interactions, signed hypergraphs enable hyperedges to connect multiple nodes simultaneously, capturing group interactions that are critical in educational settings. For example, a single hyperedge in a signed hypergraph can represent the collective response of multiple students to a specific question, providing a richer and more expressive representation of learning behaviors. This capability allows the signed hypergraph to encode non-local relationships, such as shared misconceptions among students or the interplay of multiple knowledge points within a course, which signed graphs cannot adequately capture. These characteristics highlight the potential of signed hypergraphs for more comprehensive modeling of student-content interactions and the advancement of personalized learning systems.

## 3. Method

### 3.1. Framework Overview

As shown in Figure 1, EduLLM consists of three key modules: i) an LLM-based semantic extraction module, which extracts key components from MCQs (e.g., stems, options, answers, and explanations) and generates semantic embeddings enriched with contextual knowledge; ii) a signed hypergraph construction module, which models interactions between students and MCQs by representing students and questions as nodes and encoding correct and incorrect responses as positive and negative signed hyperedges; and iii) FraS-HNN, a framelet-based signed hypergraph neural network equipped with low-pass and high-pass filters for multi-frequency hypergraph learning. As the novel component of the framework, FraS-HNN is detailed in the following section, while further details about the first two components are provided in the **Appendix D**.

### 3.2. Construction of FraS-HNN

We propose FraS-HNN to analyze signed hypergraphs by decomposing their signals into multi-frequency components. This approach enables effective feature learning by leveraging framelet theory, capturing both shared (low-pass) and individual (high-pass) features within the signed hypergraph.

**Framelet Construction on Signed Hypergraphs.** Let $\mathcal{G} = (\mathcal{V}, \mathcal{E})$ denote a signed hypergraph, where $\mathcal{V}$ represents the set of $N$ nodes, and $\mathcal{E}$ represents the signed hyperedges (including both positive and negative ones). The signed hypergraph Laplacian $\mathcal{L}_s$ encodes the structural relationships within $\mathcal{G}$. Its eigendecomposition is expressed as: $\mathcal{L}_s = \mathbf{M}\mathbf{\Delta}\mathbf{M}^\top$, where $\mathbf{M} = [\mathbf{m}_1, \mathbf{m}_2, \ldots, \mathbf{m}_N]$ is the matrix of eigenvectors, and $\mathbf{\Delta} = \text{diag}(\delta_1, \delta_2, \ldots, \delta_N)$ contains the eigenvalues.

Framelets are constructed using scaling and wavelet functions $\Gamma = \{\varphi; \psi^{(1)}, \ldots, \psi^{(k)}\}$, with associated filter banks

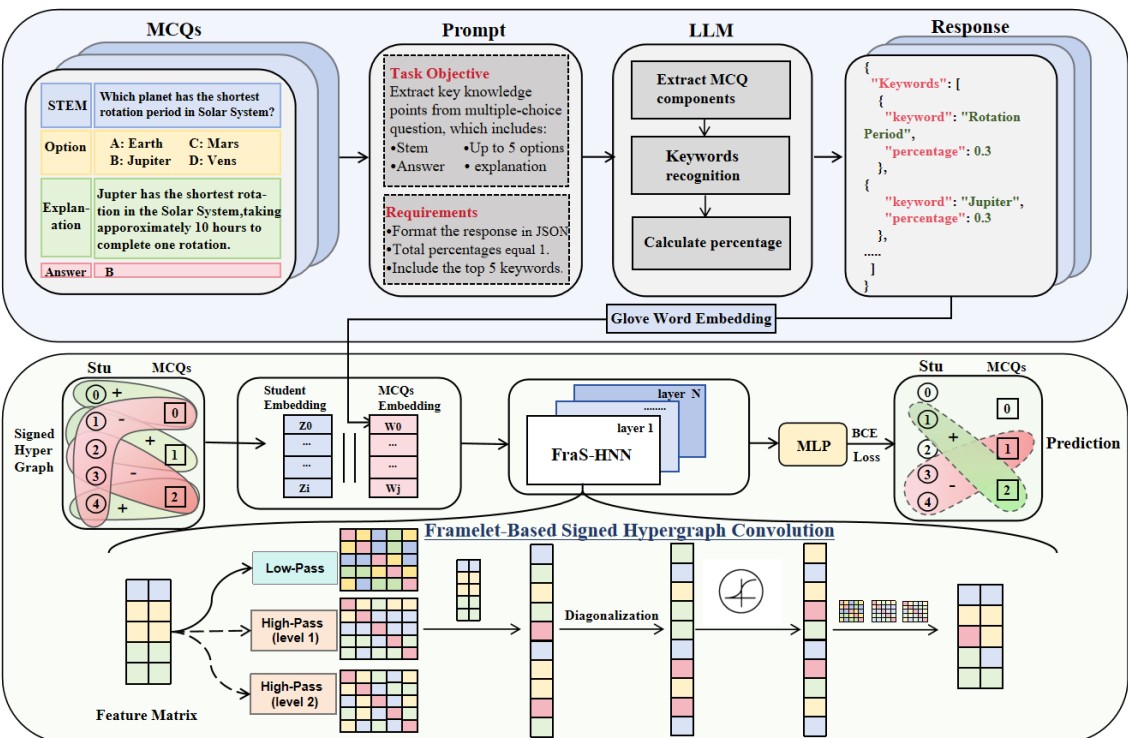

*Figure 1.* Schematic of our proposed EduLLM framework: The process begins with extracting semantic information from MCQs using an LLM, where key components such as the question stem, options, answers, and explanations are identified and formatted into a structured representation. The LLM generates semantic embeddings for the MCQs by recognizing keywords and calculating their relative contributions. These embeddings, combined with Glove word embeddings, are incorporated into a signed hypergraph structure that models the interactions between students and MCQs. In this signed hypergraph, students and MCQs are represented as nodes, and their interactions are encoded as positive or negative signed hyperedges, reflecting correct and incorrect responses, respectively. FraS-HNN, the proposed framelet-based signed hypergraph neural network, processes this hypergraph using multi-frequency signal analysis. Specifically, the proposed spectral-based signed hypergraph convolution module leverages framelet-based low-pass and high-pass filters to capture shared patterns among nodes while highlighting individualized distinctions within the hypergraph structure. These processed features are then propagated through multiple layers of FraS-HNN to learn comprehensive embeddings for students and MCQs. The final prediction module employs a MLP with a binary cross-entropy loss function to predict student performance, completing the framework.

$\Theta = \{g; h^{(1)}, \ldots, h^{(k)}\}$. These filters satisfy:

$$\widehat{\varphi}(2\omega) = \widehat{g}(\omega)\widehat{\varphi}(\omega), \quad \widehat{\psi^{(r)}}(2\omega) = \widehat{h^{(r)}}(\omega)\widehat{\varphi}(\omega), \quad (2)$$

where $\widehat{f}(\omega)$ is the Fourier transform of $f$. The scaling function $\varphi$ captures shared patterns (low-pass), while the wavelet functions $\psi^{(r)}$ ($r = 1, \ldots, k$) highlight distinctions (high-pass). For a node $v$ at scale $j$, the low-pass and high-pass framelets are defined as:

$$\Phi_{j,v}(u) = \sum_{p=1}^{N} \widehat{\varphi}\left(\frac{\delta_p}{2^j}\right) m_p(v) m_p(u), \quad (3)$$

$$\Psi_{j,v}^{(r)}(u) = \sum_{p=1}^{N} \widehat{\psi^{(r)}}\left(\frac{\delta_p}{2^j}\right) m_p(v) m_p(u). \quad (4)$$

**Filter Design with Haar-Type Functions.** For computational efficiency, FraS-HNN employs Haar-type filters to

construct the framelets. For a system with two levels of decomposition ($j = 1, 2$) and one high-pass filter ($r = 1$), the filters are defined as:

$$\widehat{\varphi}\left(\frac{\Delta}{2}\right) = \cos\left(\frac{\Delta}{8}\right)\cos\left(\frac{\Delta}{16}\right), \quad (5)$$

$$\widehat{\psi^{(1)}}\left(\frac{\Delta}{2}\right) = \sin\left(\frac{\Delta}{8}\right)\cos\left(\frac{\Delta}{16}\right), \quad (6)$$

$$\widehat{\psi^{(1)}}\left(\frac{\Delta}{4}\right) = \sin\left(\frac{\Delta}{16}\right), \quad (7)$$

yielding one low-pass and two high-pass components. These filters enable efficient decomposition of signals into multi-frequency components.

**Framelet-Based Signed Hypergraph Convolution.** From the implementation perspective, eigendecomposition of $\mathcal{L}_s$ can be computationally expensive for large

hypergraphs. FraS-HNN addresses this by approximating spectral filters using Chebyshev polynomials:

$$\widehat{\varphi}(\mathbf{\Delta}) \approx \sum_{n=0}^{T} c_n \mathcal{T}_n(\mathbf{\Delta}), \ \widehat{\psi^{(r)}}(\mathbf{\Delta}) \approx \sum_{n=0}^{T} d_n^{(r)} \mathcal{T}_n(\mathbf{\Delta}), \ (8)$$

where $\mathcal{T}_n(x)$ is the $n$-th Chebyshev polynomial. This approximation reduces computational costs while retaining spectral properties.

The framelet-based signed hypergraph convolution operator is formally defined as:

$$\mathcal{F}(\mathbf{X}) = \sum_{j=1}^{J} \sum_{r=1}^{k} \mathcal{T}_{j,r}^{\top} \text{diag}(\theta_{j,r}) \mathcal{T}_{j,r} \mathbf{X} \\ + \mathcal{T}_{0,J}^{\top} \text{diag}(\theta_{0,J}) \mathcal{T}_{0,J} \mathbf{X}, \quad (9)$$

where: $\mathcal{T}_{0,J}$ and $\mathcal{T}_{j,r}$ are the low-pass and high-pass decomposition operators, respectively, $\theta_{j,r}$ and $\theta_{0,J}$ are trainable spectral filters applied to the corresponding frequency components. The decomposition operators are defined as:

$$\mathcal{T}_{0,J} = \mathbf{M}^{\top} \widehat{\varphi} \left( \frac{\mathbf{\Delta}}{2^J} \right) \mathbf{M}, \mathcal{T}_{j,r} = \mathbf{M}^{\top} \widehat{\psi^{(r)}} \left( \frac{\mathbf{\Delta}}{2^{j+1}} \right) \mathbf{M}. \ (10)$$

These operators project the signal $\mathbf{X}$ onto the low-pass and high-pass framelets, enabling multi-frequency analysis.

**Building FraS-HNN.** FraS-HNN is constructed by stacking multiple layers of framelet-based signed hypergraph convolution. To improve learning stability and mitigate the over-smoothing issue, FraS-HNN employs initial residual and identity mapping techniques introduced in (Chen et al., 2020). Each layer is defined as:

$$\mathbf{X}^{(\ell+1)} = \sigma \bigg( \Big( (1 - \alpha_\ell) \mathcal{F}(\mathbf{X}^{(\ell)}) + \alpha_\ell \mathbf{X}^{(0)} \Big) \\ \cdot \Big( (1 - \beta_\ell) \mathbf{I} + \beta_\ell \mathbf{\Theta}^{(\ell)} \Big) \bigg), \quad (11)$$

where $\mathbf{X}^{(\ell)}$ is the feature matrix at layer $\ell$, $\mathbf{X}^{(0)}$ is the initial feature matrix, $\mathcal{F}(\cdot)$ represents the framelet-based signed hypergraph convolution operator, $\mathbf{\Theta}^{(\ell)}$ is a trainable diagonal scaling matrix, $\alpha_\ell \in [0, 1]$ balances the contributions of the initial and current features, $\beta_\ell \in [0, 1]$ adjusts the impact of the identity mapping, $\sigma(\cdot)$ is a non-linear activation function (e.g., ReLU).

### 3.3. Training Objective

After obtaining the embeddings for each student $i$ and question $j$ from the signed hypergraph learning module equipped with the FraS-HNN model, we represent these embeddings as $\mathbf{z}_{u_i}$ and $\mathbf{w}_j \in \mathbb{R}^d$, where $d$ denotes the embedding dimension. The function $f(\mathbf{z}_{u_i}, \mathbf{w}_j) \rightarrow \{-1, +1\}$ is used

to predict the sign of the unobserved hyperedge $e_{ij}$. To accomplish this, we concatenate the student embedding $\mathbf{z}_{u_i}$ with with the question embedding $w_j$, forming a combined vector. This concatenated vector is then passed through a multi-layer perceptron (MLP) to predict the edge sign, as described by the following form:

$$y_{pred} = \text{MLP}(\mathbf{z}_{u_i} \| \mathbf{w}_j). \quad (12)$$

Here, the predicted score $y_{pred}$ corresponds to the likelihood that the hyperedge sign is positive, with higher scores indicating a stronger probability of a positive sign, and lower scores suggesting a negative sign.

For the training objective, we adopt the binary cross-entropy loss function to optimize the hyperedge sign prediction task, that is:

$$\mathcal{L}_{CE} = -y \log(y_{pred}) + (1 - y) \log(1 - y_{pred}), \quad (13)$$

where $y$ denotes the ground-truth label of the hyperedge sign, with mapping $\{-1, 1\}$ to $\{0, 1\}$.

Additional theoretical properties of FraS-HNN, along with rigorous proofs, are provided in the **Appendix B**.

## 4. Experiments

In this section, we conduct a series of experiments to evaluate the effectiveness of the proposed EduLLM framework. The primary objectives are to address the following research questions:

- **Q1**: How does EduLLM perform compared to state-of-the-art signed graph representation learning methods and classic graph neural network models?

- **Q2**: What are the contributions of high-pass information, low-pass information, and semantic feature embeddings to EduLLM's performance?

- **Q3**: How do key hyperparameters impact the performance of EduLLM?

- **Q4**: How robust is the EduLLM model with respect to the LLM-induced semantic representation embeddings?

- **Q5**: How does the proposed FraS-HNN module benefit the EduLLM framework from a hypergraph learning perspective?

### 4.1. Baselines and Evaluation Metrics

To address **Q1**, we evaluate EduLLM on five real-world datasets by comparing its performance against several baselines, including Random Embedding, Graph Convolutional Network (GCN) (Kipf & Welling, 2017), Graph Attention

Networks (GAT) (Veličković et al., 2018), Signed Graph Convolutional Network (SGCN) (Derr et al., 2018), Signed Bipartite Graph Neural Network (SBGNN) (Huang et al., 2021), Signed Bipartite Graph Contrastive Learning (SBCL) (Wang et al., 2024b), and its variant version LLM-SBCL (Wang et al., 2024b), see more details in the **Appendix E.1**. For **Q2** and **Q3**, we conduct ablation studies and sensitivity analyses to examine the contributions of individual components and the influence of key hyperparameters. To answer **Q4**, we perform robustness tests to assess the reliability and applicability of the integrated large language models. For **Q5**, we replace the FraS-HNN module with other existing HNN models while keeping the remaining components of EduLLM unchanged. This allows for a convincing evaluation of the role that FraS-HNN plays in learning signed hypergraphs within the EduLLM framework.

Given the imbalance between positive and negative links in the datasets, we employ two evaluation metrics: area under the curve (AUC) and binary-average F1 score (Binary-F1). Both metrics provide a balanced assessment of performance, with higher values indicating better results.

### 4.2. Datasetes

Multiple Choice Questions (MCQs) are a fundamental tool for assessing students' knowledge and learning progress, particularly on online education platforms where they play a critical role in evaluating learners. In this study, we utilize real-world datasets containing interaction information between students and MCQs. Each interaction is recorded as either a correct response (marked as '+') or an incorrect response (marked as '-'), enabling the construction of both signed graphs (to align with baseline models) and signed hypergraphs (to fit the problem formulation of EduLLM). The five datasets (Wang et al., 2024b) used in our experiments are derived from courses at three universities: the *Biology* and *Law* courses from the University of Auckland, the *Cardiff20102* Medical School course from Cardiff University, and the biochemistry courses *Sydney19351* and *Sydney23146* from the University of Sydney.

**Signed Hypergraph Construction.** To construct the signed hypergraph for each dataset, we leverage the students' response records. Each student's response to a question is classified as either correct or incorrect. Since answering processes for questions are independent, the hypergraph structure naturally emerges from the data. Specifically: i) For each question node $q_j \in \mathcal{V}$, students who correctly answer the question form a positive hyperedge ('+1') connecting the question node with the corresponding student nodes $u_i \in \mathcal{V}$; ii) Similarly, students who answer the question incorrectly form a negative hyperedge ('-1') with the same question node. Thus, each dataset is represented as a signed hypergraph, where student nodes and question nodes

*Table 1.* Statistics of the five real-world datasets.

|  | Biology | Law | Cardiff | Sydney19 | Sydney23 |
|---|---|---|---|---|---|
| $|\mathcal{U}|$ | 761 | 528 | 383 | 382 | 198 |
| $|\mathcal{V}|$ | 380 | 5600 | 1171 | 457 | 748 |
| $|\mathcal{E}|$ | 760 | 11,200 | 2,342 | 914 | 1,496 |
| $|E|$ | 76,613 | 88,563 | 64,524 | 24,032 | 24,050 |
| Pos Link | 66.5% | 93.1% | 60.0% | 53.1% | 70.6% |
| Neg Link | 33.5% | 6.9% | 40.0% | 46.9% | 29.4% |

interact through hyperedges labeled to reflect the correctness of responses. The hypergraph captures the higher-order relationships between multiple students and questions, providing a richer structural representation compared to the graph-based and signed graph-based models.

Table 1 provides a detailed comparison of the data characteristics for both signed hypergraphs and signed graphs constructed from the five real-world datasets. In this table, the number of hyperedges within the signed hypergraph is denoted by $|\mathcal{E}|$, which captures the higher-order relationships between multiple students and a single question. In contrast, the number of edges within the signed graph is denoted by $|E|$, representing the direct, individual interactions between students and questions. Naturally, $|\mathcal{E}|$ is significantly smaller than $|E|$, reflecting the ability of hypergraphs to represent co-occurrence (higher-order) relationships among students and questions.

Further details on the datasets and the process of formulating the original 'students-MCQs' data sources as signed hypergraphs are provided in **Appendix C**.

### 4.3. Experimental Setup

We note that, in our experiments, the semantic embeddings are pre-encoded and provided by (Wang et al., 2024b), rather than being directly derived from the original multiple-choice question (MCQ) data (e.g., question stems, options, and explanations). As a result, it is not feasible to directly compare the performance of EduLLM using different LLM modules. To roughly investigate the impact of LLM-induced embeddings, we conduct a robustness study as described in Section 4.7 (for Q4). Meanwhile, using the same pre-encoded semantic embeddings ensures a fair comparison when analyzing the specific contribution of the FraS-HNN module, as presented in Section 4.8 (for Q5).

The datasets are partitioned into three subsets to ensure a robust evaluation: 85% for training to capture underlying data patterns, 5% for validation to fine-tune hyperparameters, and 10% for testing to assess the model's generalization capability. Each model was trained for 300 epochs to ensure convergence while reducing the risk of overfitting. To enhance the reliability of the results, all experiments were repeated ten times. The mean and standard deviation of the

*Table 2.* Performance comparison on five real-world educational datasets (average binary-F1 ± standard deviation).

|  | Biology | Law | Cardiff20102 | Sydney19351 | Sydney23146 |
|---|---|---|---|---|---|
| Random | $0.350 \pm 0.010$ | $0.472 \pm 0.001$ | $0.136 \pm 0.062$ | $0.290 \pm 0.014$ | $0.288 \pm 0.035$ |
| GCN | $0.682 \pm 0.058$ | $0.823 \pm 0.010$ | $0.677 \pm 0.024$ | $0.642 \pm 0.021$ | $0.728 \pm 0.013$ |
| GAT | $0.618 \pm 0.013$ | $0.817 \pm 0.050$ | $0.571 \pm 0.013$ | $0.564 \pm 0.022$ | $0.608 \pm 0.020$ |
| SGCN | $0.768 \pm 0.040$ | $0.840 \pm 0.013$ | $0.607 \pm 0.033$ | $0.635 \pm 0.044$ | $0.726 \pm 0.040$ |
| SBGNN | $0.753 \pm 0.014$ | $0.861 \pm 0.034$ | $0.712 \pm 0.016$ | $0.673 \pm 0.016$ | $0.712 \pm 0.021$ |
| SBCL | $0.772 \pm 0.016$ | $0.901 \pm 0.016$ | $0.718 \pm 0.018$ | $0.674 \pm 0.021$ | $0.733 \pm 0.019$ |
| LLM-SBCL | $\underline{0.787 \pm 0.014}$ | $\underline{0.908 \pm 0.018}$ | $\underline{0.734 \pm 0.023}$ | $\underline{0.694 \pm 0.021}$ | $\underline{0.760 \pm 0.022}$ |
| EduLLM (Ours) | $\mathbf{0.809 \pm 0.010}$ | $\mathbf{0.945 \pm 0.005}$ | $\mathbf{0.753 \pm 0.011}$ | $\mathbf{0.712 \pm 0.016}$ | $\mathbf{0.829 \pm 0.006}$ |

performance metrics were reported to account for variations due to data splits and random initialization, thereby ensuring robust and unbiased evaluation. Detailed reproducibility parameters for the optimal results obtained in our experiments are provided in the **Appendix E.2**.

### 4.4. Results and Discussion (Q1)

To address **Q1**, we compare EduLLM with various baseline models on the five datasets. The comparison results, summarized in Table 2, lead to the following key observations:

**Superior Performance of EduLLM.** Across all datasets, EduLLM consistently outperformed all baseline models, achieving the highest binary F1 scores. This result underscores the advantages of hypergraph-based models in capturing multi-scale signed information. The use of framelet-based hypergraph convolution, which effectively integrates low-pass and high-pass information, was instrumental in achieving this superior performance.

**Limitations of Vanilla GNN Models.** As evident from Table 2, traditional unsigned graph learning methods such as GCN and GAT demonstrate limited effectiveness in modeling signed graphs or hypergraphs. These models are unable to fully capture the intricate signed interactions inherent in educational contexts. In contrast, EduLLM leverages its signed hypergraph structure to model these complex relationships, consistently delivering better predictive performance across all datasets.

**FraS-HNN vs. Signed GNNs.** Specifically, EduLLM also outperforms state-of-the-art signed graph models, including SGCN, SBGNN, SBCL, and its variant LLM-SBCL, which integrates large language models. This significant improvement highlights the strength of FraS-HNN's framelet-based hypergraph convolution, which effectively captures both local and global signed information. This enhanced higher-order feature expressiveness give EduLLM a distinct advantage over these competing models.

**Robustness and Stability.** Relatively, EduLLM exhibits low standard deviations across all datasets, reflecting strong stability and generalization capability. Moreover, the model maintains solid performance despite the diversity of the

datasets, demonstrating its potential adaptability and applicability in varying educational contexts.

### 4.5. Ablation Study (Q2)

To address **Q2**, we conduct an ablation study to evaluate the contributions of high-pass information, low-pass information, and semantic feature embeddings to the performance of the EduLLM model. In particular, we compare the full model with three ablated variants: i) **w/o High**: This variant excludes the **high-pass filter**, retaining only the low-pass filter. It evaluates the role of high-frequency information in capturing fine-grained distinctions within the hypergraph; ii)**w/o Low**: This variant removes the **low-pass filter**, relying solely on the high-pass filter. It assesses the contribution of low-frequency information in modeling shared preferences and global patterns; iii) **w/o LLM**: This variant eliminates the **semantic embeddings** derived from the LLM and relies solely on structural embeddings. It investigates the significance of incorporating semantic information into the hypergraph representation.

**Observation and Discussion.** EduLLM's integration of both low-pass and high-pass features is theoretically motivated, as discussed in the **Appendix B**, where FraS-HNN is designed to comprehensively capture the diversity and complexity of signed hypergraph signals. This balanced integration of high- and low-frequency information, combined with semantic embeddings, forms a robust foundation for modeling diverse patterns within the data. The results of the ablation study, presented in Table 3, demonstrate the substantial contributions of each component to the overall performance of EduLLM. Removing any of the key components leads to a significant drop in binary-F1 scores across all datasets, underscoring their importance. Specifically, the high-pass component plays a vital role in capturing fine-grained distinctions, while the low-pass component effectively models global relationships. Together, these components of the framelet-based hypergraph convolution highlight the potential for designing more advanced hypergraph neural networks for signed hypergraph learning. Furthermore, the semantic embeddings induced by the LLM significantly enhance the representation of knowledge

Table 3. Ablation study results (average binary-F1 ± standard deviation).

|  | Full Model | w/o High | w/o Low | w/o LLM |
|---|---|---|---|---|
| Biology | **0.809 ± 0.010** | 0.801 ± 0.005 | 0.799 ± 0.005 | 0.795 ± 0.017 |
| Law | **0.945 ± 0.005** | 0.943 ± 0.017 | 0.934 ± 0.029 | 0.941 ± 0.013 |
| Cardiff | **0.753 ± 0.011** | 0.731 ± 0.019 | 0.741 ± 0.017 | 0.740 ± 0.004 |
| Sydney19 | **0.712 ± 0.016** | 0.705 ± 0.014 | 0.693 ± 0.006 | 0.703 ± 0.033 |
| Sydney23 | **0.829 ± 0.006** | 0.817 ± 0.261 | 0.823 ± 0.012 | 0.819 ± 0.016 |

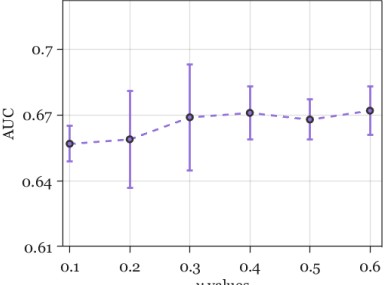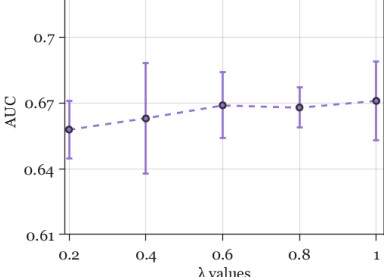

Figure 2. Results for parameter sensitivity analysis on the Law dataset: $\alpha$ (LHS), $\gamma$ (middle), $\lambda$ (RHS), respectively.

points from the MCQs, demonstrating their ability to complement the structural embeddings. Overall, these findings highlight the individual and collective contributions of the high-/low-pass components, and the LLM-induced semantic information, to the overall merits of EduLLM.

### 4.6. Parameter Sensitivity Analysis (Q3)

To address **Q3**, we perform a detailed sensitivity analysis on three key hyperparameters: $\alpha$, $\gamma$, and $\lambda$. These hyperparameters are integral to balancing various components of the model and influence its performance: 1) $\alpha$ regulates the balance between the current feature representation and the initial feature map; 2) $\gamma$ controls the relative contribution of the two computational branches in the model: one based on random filtering and the other on adjacency matrix propagation. It adjusts the interaction dynamics between these two pathways; 3) $\lambda$ modulates the weight of each layer's feature updates relative to the initial features through the scaling parameter $\theta$. It affects the overall hierarchical learning of feature representations across layers.

To evaluate the impact of these hyperparameters, we conduct experiments by varying their values over a predefined range. Due to page limitations, we present the sensitivity analysis results on the Law dataset in Figure 2, while additional results on other datasets are included in the **Appendix E.4**. As shown clearly in Figure 2, EduLLM performs stably across a broad range of values for all three hyperparameters. While slight variations in performance are observed as parameter values change, the overall trends remain consistent, and no abrupt performance degradation occurs. This stability, to a certain extent, highlights the robustness of EduLLM to parameter tuning, ensuring its adaptability and effectiveness across different configurations.

### 4.7. Further Assessment for the Role of LLM-induced Semantic Representation (Q4)

To address **Q4**, we conduct additional studies to evaluate the robustness of EduLLM with respect to the semantic representations induced by the large language model. Due to the limitation that the semantic embeddings in our experiments are pre-encoded and provided by Wang et al. (Wang et al., 2024b), rather than derived directly from the original MCQ data (e.g., question stems, options, and explanations), it is not feasible to directly compare EduLLM using different LLMs. As an alternative, we simulate variations in semantic embeddings by introducing different levels of noise to the pre-encoded embeddings of the MCQs. This approach serves as a simple substitution for testing the robustness of the LLM backbone, as the variance caused by the noise approximates the differences in embeddings generated by different LLMs. Specifically, we add additive white Gaussian noise $\mathcal{M} \sim \mathcal{N}(0, \sigma^2) \in \mathbb{R}^d$ to the semantic embedding tensor $\mathcal{X} \in \mathbb{R}^d$ The noise level $\sigma$ $\sigma = p(\max(\mathcal{X}) - \min(\mathcal{X}))$, where $p \in \{0.01, 0.03, 0.05, 0.08, 0.1\}$ indicates indicates the relative strength of the added noise.

Following this design, we comprehensively evaluate the predictive performance of EduLLM under different noise levels. The results, presented in Fig. 3, show that as $p$ increases, the model's performance remains relatively stable, with only minor declines observed at higher noise levels. This smooth degradation is expected and indicates that EduLLM is robust to variations in the semantic embeddings, suggesting that replacing the current LLM backbone with a different one does not significantly impact the overall task performance. We should note that these experiments validate the robustness and reliability of EduLLM in handling noise interference, highlighting its ability to maintain consistent predictive ac-

*Table 4.* Performance comparison of EduLLM variants with different HNN modules.

| | Sydney19351 | Sydney23146 | Biology | Cardiff20102 | Law |
|---|---|---|---|---|---|
| EduLLM with HGNN | $0.606 \pm 0.014$ | $0.619 \pm 0.013$ | $0.673 \pm 0.006$ | $0.624 \pm 0.007$ | $0.905 \pm 0.011$ |
| EduLLM with HyperGCN | $0.620 \pm 0.012$ | $0.650 \pm 0.038$ | $0.651 \pm 0.016$ | $0.625 \pm 0.023$ | $0.901 \pm 0.032$ |
| EduLLM with AllDeepSets | $0.626 \pm 0.017$ | $0.660 \pm 0.012$ | $0.697 \pm 0.008$ | $0.637 \pm 0.023$ | $0.898 \pm 0.009$ |
| EduLLM with AllSetTransformer | $0.618 \pm 0.022$ | $0.661 \pm 0.010$ | $0.689 \pm 0.016$ | $0.644 \pm 0.029$ | $0.906 \pm 0.009$ |
| EduLLM with ED-HNN | $0.662 \pm 0.023$ | $0.708 \pm 0.032$ | $0.715 \pm 0.039$ | $0.673 \pm 0.026$ | $0.910 \pm 0.011$ |
| EduLLM with SheafHyperGNN | $0.684 \pm 0.023$ | $0.711 \pm 0.030$ | $0.732 \pm 0.022$ | $0.687 \pm 0.025$ | $0.914 \pm 0.013$ |
| EduLLM with FraS-HNN | $\mathbf{0.712 \pm 0.016}$ | $\mathbf{0.829 \pm 0.006}$ | $\mathbf{0.809 \pm 0.010}$ | $\mathbf{0.753 \pm 0.011}$ | $\mathbf{0.945 \pm 0.005}$ |

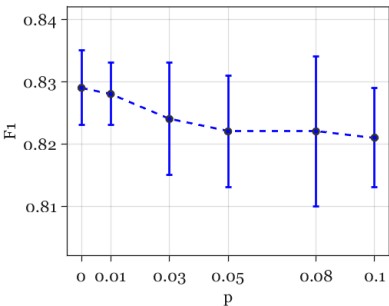

*Figure 3.* Performance demonstration for robustness analysis on the Sydney23 dataset.

curacy even when faced with perturbations in the semantic representations. This robustness underscores the potential of EduLLM to generalize across different LLM backbones and adapt to varying sources of semantic information. However, as a limitation, this empirical study may not fully reflect the role of LLM-induced semantic representations in EduLLM.

### 4.8. Further Study on the Role of FraS-HNN

In order to verify the advantages of FraS-HNN that benefit EduLLM framework, we have included comparisons with several HNN baselines, including HGNN (Feng et al., 2019), HyperGCN (Yadati et al., 2019), AllDeepSets (Chien et al., 2022), AllSetTransformer (Chien et al., 2022), ED-HNN (Wang et al., 2023), and SheafHyperGNN (Duta et al., 2023), by replacing the FraS-HNN backbone of EduLLM with each of these HNN modules. As shown in Table 4, EduLLM (with FraS-HNN) consistently outperforms the variants equipped with each HNN module across all datasets, demonstrating the effectiveness of FraS-HNN in modeling signed high-order interactions and its potential to benefit future research in (signed) hypergraph learning.

### 5. Related Works

Hypergraph neural networks and hypergraph representation learning have emerged as powerful tools for capturing higher-order correlations in complex data, thereby extending the capabilities of traditional graph neural networks (Antelmi et al., 2023; Kim et al., 2024). By modeling relationships among multiple entities through hyperedges,

HNNs are well-suited for various real-world applications, including recommendation systems, social network analysis, and bioinformatics. However, despite their promise, existing representative HNNs, such as HGNN (Feng et al., 2019), HyperGCN (Yadati et al., 2019), HNHN (Dong et al., 2020), HCHA (Bai et al., 2021), HCoN (Wu et al., 2023), UniGNN (Huang & Yang, 2021), AllDeepSets and AllSet-Transformer (Chien et al., 2022), ED-HNN (Wang et al., 2023), SheafHyperGCN and SheafHyperGNN (Duta et al., 2023), are primarily limited to undirected and unsigned hypergraph scenarios. Efforts to design HNNs for signed hypergraphs remain in the early stages. Meanwhile, the rapid development of LLMs has opened new opportunities to combine hypergraph learning with LLMs, particularly in applications where learning from raw data is challenging due to the multimodal nature of the information and the complex, non-Euclidean relationships inherent in the data. LLMs are well-suited to handle the multimodal aspect, while hypergraphs offer an effective framework for modeling non-Euclidean relationships. To the best of our knowledge, our work is the first to combine LLMs and signed hypergraph learning for student performance prediction in a real-world educational context. An extended version of the related works is provided in the **Appendix A**.

### 6. Conclusion

This paper proposes EduLLM, a framework that integrates LLMs with framelet-based signed hypergraph neural networks (FraS-HNN) for student performance prediction. As a novel spectral-based model for signed hypergraph learning, FraS-HNN leverages collaborative low-pass and high-pass filters to enable multi-frequency feature learning, with rigorous theoretical guarantees for its multi-frequency analysis properties in both forward and inverse transforms. This design effectively captures both shared patterns among students and individual distinctions in their learning behaviors. By integrating LLM-induced semantic embeddings with this advanced hypergraph representation, EduLLM achieves superior performance across multiple datasets, addressing the structural and semantic complexities of personalized learning. For future work, we plan to develop more advanced signed hypergraph neural networks and explore their applications in various domains.

## Acknowledgments

This work was supported in part by the "Pioneer" and "Leading Goose" R&D Program of Zhejiang (No. 2024C03262), and by the National Natural Science Foundation of China (Nos. U21A20473, 62172370). L. Bai acknowledges support from the National Natural Science Foundation of China (No. T2122020). F. Cao acknowledges support from the National Natural Science Foundation of China (Nos. 62176244, 62032022). K. Lu acknowledges support from the National Natural Science Foundation of China (Nos. U23A20388, 62320106007).

## Impact Statement

This paper aims to advance the field of Graph Machine Learning, focusing on key topics such as hypergraph/graph neural networks, hypergraph/graph representation learning, spectral-based hypergraph/graph convolution, and their advanced applications in intelligent education. There may be certain societal consequences of our work, none of which we feel must be specifically highlighted here.

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

# A. Extended Related Works

## A.1. Student Performance Prediction

Student performance prediction involves leveraging historical data and advanced analytical methods to forecast a student's future performance in academic tasks such as exams, assignments, and course completion. The primary objective is to predict students' performance based on their prior learning data, enabling educational institutions and instructors to tailor support and guidance to individual learning needs (Lin et al., 2023).

Early models for student performance prediction relied heavily on traditional machine learning methods, such as logistic regression, decision trees, and support vector machines. However, these approaches faced significant limitations, particularly in their reliance on manually extracted features and their inability to capture the complexity of student behavior. With the rise of deep learning, classic models such as Convolutional Neural Networks (CNN), Recurrent Neural Networks (RNN), Long Short-Term Memory (LSTM) networks, and Deep Neural Networks (DNN) have gained prominence due to their ability to extract more complex features (Lin et al., 2023). Specifically, efforts to enhance personalized learning have resulted in two main categories of prediction models: static models and sequential models (Li et al., 2020; Thaker et al., 2019). Static models, such as those studied by Wei et al. (2020) and Daud et al. (2017), rely solely on historical data for predicting future performance. While effective, these models fail to account for temporal dependencies and sequential relationships within the learning process. On the other hand, sequential models like knowledge tracing and its variants (Piech et al., 2015; Nakagawa et al., 2019) model the evolving knowledge of students, allowing for dynamic updates to their knowledge base and providing more accurate predictions in certain settings.

However, despite these advances, many existing methods still overlook the complex, non-linear interactions among students and learning content, as well as the rich semantic information embedded in educational resources. For instance, many models focus on analyzing temporal dependencies or basic student-resource interactions without considering how the deeper relationships between different learning materials influence student performance. In contrast, our proposed EduLLM framework not only leverages the content of the questions and learner-question interaction data, but also integrates semantic embeddings from LLMs, capturing the rich semantic context of educational resources. By using both structural and semantic features, EduLLM overcomes the limitations of traditional methods, improving prediction accuracy and providing more nuanced insights into student learning behaviors.

## A.2. Signed Graph Neural Networks

Student performance prediction can be framed as a link sign prediction task within a signed bipartite graph, where positive edges indicate correct answers and negative edges indicate incorrect ones (Derr et al., 2018; Zhang et al., 2023). Early methods for signed graph embedding, such as SIDE (Kim et al., 2018) and SGDN (Jung et al., 2020), used random walks to generate node embeddings. While these methods successfully captured structural information, they struggled to model the nuanced relationships between the signs. Other techniques, such as signed Laplacian embedding (Derr et al., 2018) and matrix factorization (Zhang et al., 2023), utilized mathematical tools to extract structural information from signed graphs but were limited in handling the inherent balance and contrast of signed edges.

Recent advancements have incorporated neural network-based approaches to address these limitations. For instance, SGCN (Derr et al., 2018) extends GCNs (Kipf & Welling, 2017) by incorporating balance theory to optimize sign prediction. SiNE (Kim et al., 2018) combines triangular structures with balance theory to enhance signed graph representation learning. While these models made strides in capturing local structural information, they still fall short of exploiting the global features of signed graphs. More recent approaches, inspired by contrastive learning, such as Signed Graph Contrastive Learning (SGCL) (Shu et al., 2021), SBGNN (Huang et al., 2021), and Signed Bipartite Graph Contrastive Learning (SBCL) (Wang et al., 2024b), have focused on learning from signed graphs and signed bipartite graphs. However, these models primarily focus on extracting local structural information, addressing only pairwise relationships between nodes. They do not consider higher-order relationships, such as co-occurrences or interactions among multiple nodes, which are critical in more complex scenarios. In contrast, our proposed EduLLM framework utilizes signed hypergraph models, which extend traditional graph structures by considering higher-order interactions through hyperedges. This allows EduLLM to capture richer, more complex relationships, addressing both local and global patterns within the data.

## A.3. LLMs for Education

LLMs have revolutionized several areas of the education sector, particularly in personalized learning, automated assessment, and educational assistance (Kasneci et al., 2023; Wang et al., 2024c). For example, Brown et al. (2020) demonstrate the potential of GPT-3 in few-shot learning, highlighting its ability to personalize learning by analyzing students' behaviors and language expressions. These models can generate individualized learning paths and resources, improving learning outcomes. However, practical applications face certain challenges, particularly the need for large amounts of labeled data for fine-tuning, which is often scarce and imbalanced in educational settings, limiting the models' generalization and deployment. The GPT-4 technical report (Achiam et al., 2023) demonstrates that LLMs can perform at a student level on standardized tests in various mathematics subjects, including physics and computer science, for both multiple-choice and open-ended questions. Furthermore, studies have shown that LLMs can be effectively used as writing or reading assistants in educational contexts. For instance, a recent study (Susnjak & McIntosh, 2024) finds that ChatGPT is capable of generating coherent and logically consistent responses across a range of disciplines, providing both depth and breadth in its answers. Another quantitative analysis (Malinka et al., 2023) indicate that students who use ChatGPT, refining or incorporating the model's outputs into their own answers, outperform their peers in certain computer security courses. Additionally, recent reviews (Tan et al., 2023; Wang et al., 2024c) have discussed potential applications of LLMs in educational settings, such as enhancing teacher-student collaboration, enabling personalized learning, and automating assessment processes.

While there are certain concerns associated with the use of LLMs in practical applications (Kasneci et al., 2023), such as issues related to plagiarism, potential biases in AI-generated content, overreliance on LLMs, and inequitable access for non-English speakers, LLMs have nonetheless been recognized as a powerful tool for extracting rich semantic information from multimodal educational resources. This implies that by leveraging the comprehensive semantic insights provided by LLMs, in conjunction with advanced learning models, researchers can develop more effective models capable of addressing complex tasks within specific domains, such as education. From this perspective, to the best of our knowledge, our proposed EduLLM framework is the first to combine LLMs and signed hypergraph learning for student performance prediction.

## B. Theoretical Properties of FraS-HNN

**Preliminaries and Key Notations.** Consider a signed hypergraph $\mathcal{G}_s = (\mathcal{V}, \mathcal{E})$ with $N$ nodes and signed hypergraph Laplacian $\mathcal{L}$. Let $\mathbf{M} = [\mathbf{m}_1, \mathbf{m}_2, \ldots, \mathbf{m}_N]$ denote the matrix of eigenvectors of $\mathcal{L}$, and $\mathbf{\Delta} = \mathrm{diag}(\delta_1, \delta_2, \ldots, \delta_N)$ be the diagonal matrix of the corresponding eigenvalues. Framelets over the signed hypergraph are generated by a set of scaling functions $\mathcal{S} = \{\widehat{\varphi}, \{\widehat{\psi^{(r)}}\}_{r=1}^k\} \subset L_1(\mathbb{R})$ associated with a filter bank $\eta = \{b^{(0)}; b^{(1)}, \ldots, b^{(k)}\}$, which satisfy the relations for any $\xi \in \mathbb{R}$:

$$\widehat{\varphi}(2\xi) = \widehat{b^{(0)}}(\xi)\widehat{\varphi}(\xi), \quad \widehat{\psi^{(r)}}(2\xi) = \widehat{b^{(r)}}(\xi)\widehat{\varphi}(\xi). \tag{B-1}$$

These functions are defined as follows (for clarity, Equations (3) and (4) from Section 3.2 of the main manuscript are restated here for reference):

$$\Phi_{j,v}(u) = \sum_{p=1}^N \widehat{\varphi}\left(\frac{\delta_p}{2^j}\right) m_p(v) m_p(u), \tag{B-2}$$

$$\Psi_{j,v}^{(r)}(u) = \sum_{p=1}^N \widehat{\psi^{(r)}}\left(\frac{\delta_p}{2^j}\right) m_p(v) m_p(u). \tag{B-3}$$

where $m_p(v)$ represents the $v$-th component of the eigenvector $\mathbf{m}_p$.

For integers $J, J_1$ such that $J > J_1$, we define a tight framelet system on signed hypergraphs (denoted as $\mathbf{TiFraS}(\mathcal{S}, \eta; \mathcal{G}_s)$), starting from scale $J_1$, as a non-homogeneous, stationary affine system:

$$\mathbf{TiFraS}(\mathcal{S}, \eta; \mathcal{G}_s))_{J_1}^J(\mathcal{G}_s) = \{\Phi_{J_1,v} \mid v \in \mathcal{V}\} \cup \{\Psi_{j,v}^{(r)} \mid v \in \mathcal{V}, j = J_1, \ldots, J\}_{r=1}^k. \tag{B-4}$$

**Theorem B1 (Properties of Tight Framelets on Hypergraphs).** Let $J \geq 1$ be an integer, and consider the hypergraph framelet system $\mathbf{TiFraS}_{J_1}^J(\mathcal{S}, \eta; \mathcal{G}_s)$ defined in (B-4), with hypergraph framelets $\Phi_{j,v}$ and $\Psi_{j,v}^r$. The following statements are equivalent:

*Table B-1.* Frequently used notations.

| Notation | Description |
|---|---|
| $\mathcal{G}_s = (\mathcal{V}, \mathcal{E})$ | Signed hypergraph with node set $\mathcal{V}$ and signed hyperedge set $\mathcal{E}$. |
| $\mathbf{M}, \boldsymbol{\Delta}$ | Eigenvector matrix $\mathbf{M}$ and eigenvalue diagonal matrix $\boldsymbol{\Delta}$ of the signed hypergraph Laplacian. |
| $\widehat{\varphi}(\cdot)$ | Spectral low-pass filter (scaling function). |
| $\widehat{\psi^{(r)}}(\cdot)$ | Spectral high-pass filter for the $r$-th wavelet function. |
| $\mathcal{T}_{0,J}$ | Framelet decomposition operator for low-pass components at scale $J$. |
| $\mathcal{T}_{j,r}$ | Framelet decomposition operator for the $r$-th high-pass component at scale $j$. |
| $\mathcal{F}(\mathbf{F})$ | Framelet-based signed hypergraph convolution operator. |
| $\mathbf{X}$ | Signal or feature matrix on the signed hypergraph. |
| $\mathbf{X}^{(\ell)}, \mathbf{X}^{(0)}$ | Feature matrix at layer $\ell$ and the initial feature matrix, respectively. |
| $\boldsymbol{\Theta}^{(\ell)}$ | Trainable diagonal scaling matrix at layer $\ell$. |
| $\alpha_\ell$ | Learnable parameter balancing contributions from initial features and the transformed features. |
| $\beta_\ell$ | Learnable parameter balancing the contributions of identity mapping. |
| $\sigma(\cdot)$ | Non-linear activation function (e.g., ReLU). |
| $\mathcal{T}_n(x)$ | $n$-th Chebyshev polynomial used for spectral filter approximation. |
| $c_n$ | Coefficients for approximating the low-pass filter $\widehat{\varphi}(\cdot)$ using Chebyshev polynomials. |
| $d_n^{(r)}$ | Coefficients for approximating the high-pass filter $\widehat{\psi^{(r)}}(\cdot)$ using Chebyshev polynomials. |
| $\delta_p$ | Eigenvalue associated with the $p$-th node in the signed hypergraph Laplacian. |
| $\mathcal{L}$ | Signed hypergraph Laplacian operator. |
| $\mathcal{W}_{r,j}, \mathcal{W}_{0,J}$ | Framelet decomposition matrices for high-pass $(r, j)$ and low-pass $(0, J)$ components. |
| $J$ | Number of decomposition scales (levels). |
| $k$ | Number of high-pass filters. |

(i) For each $J_1 = 1, \ldots, J$, the framelet system on hypergraphs, $\textbf{TiFraS}_{J_1}^J(\mathcal{S}, \eta; \mathcal{G}_s)$, is a tight frame for $l_2(\mathcal{G}_s)$. That is, $\forall f \in l_2(\mathcal{G}_s)$,

$$\|f\|^2 = \sum_{v \in \mathcal{V}} \left| \langle f, \Phi_{J_1,v} \rangle \right|^2 + \sum_{j=J_1}^{J} \sum_{r=1}^{v} \sum_{v \in \mathcal{V}} \left| \left\langle f, \Psi_{j,v}^{(r)} \right\rangle \right|^2. \tag{B-5}$$

(ii) For all $f \in l_2(\mathcal{G}_s)$ and for $j = 1, \ldots, J-1$, the following identities hold:

$$f = \sum_{v \in \mathcal{V}} \langle f, \Phi_{J,v} \rangle \Phi_{J,v} + \sum_{r=1}^{k} \sum_{v \in \mathcal{V}} \left\langle f, \Psi_{J,v}^{(r)} \right\rangle \Psi_{J,v}^{(r)}, \tag{B-6}$$

$$\sum_{v \in \mathcal{V}} \langle f, \Phi_{j+1,v} \rangle \Phi_{j+1,v} = \sum_{v \in \mathcal{V}} \langle f, \Phi_{j,v} \rangle \Phi_{j,v} + \sum_{r=1}^{k} \sum_{v \in \mathcal{V}} \left\langle f, \Psi_{j,v}^{(r)} \right\rangle \Psi_{j,v}^{(r)}. \tag{B-7}$$

(iii) For all $f \in l_2(\mathcal{G}_s)$ and for $j = 1, \ldots, J-1$, the following identities hold:

$$\|f\|^2 = \sum_{v \in \mathcal{V}} \left| \langle f, \Phi_{J,v} \rangle \right|^2 + \sum_{r=1}^{k} \sum_{v \in \mathcal{V}} \left| \left\langle f, \Psi_{J,v}^{(r)} \right\rangle \right|^2, \tag{B-8}$$

$$\sum_{v \in \mathcal{V}} \left| \langle f, \Phi_{j+1,v} \rangle \right|^2 = \sum_{v \in \mathcal{V}} \left| \langle f, \Phi_{j,v} \rangle \right|^2 + \sum_{r=1}^{k} \sum_{v \in \mathcal{V}} \left| \left\langle f, \Psi_{j,v}^{(r)} \right\rangle \right|^2. \tag{B-9}$$

(iv) The scaling functions in $\mathcal{S} = \{\widehat{\varphi}, \{\widehat{\psi^{(r)}}\}_{r=1}^k\} \subset L_1(\mathbb{R})$ satisfy

$$1 = \left|\widehat{\varphi}\left(\frac{\delta_q}{2^J}\right)\right|^2 + \sum_{r=1}^k \left|\widehat{\psi^{(r)}}\left(\frac{\delta_q}{2^J}\right)\right|^2 \quad \forall q = 1, \ldots, N, \tag{B-10}$$

$$\left|\widehat{\varphi}\left(\frac{\lambda_q}{2^{j+1}}\right)\right|^2 = \left|\widehat{\varphi}\left(\frac{\delta_q}{2^j}\right)\right|^2 + \sum_{r=1}^k \left|\widehat{\psi^{(r)}}\left(\frac{\delta_q}{2^j}\right)\right|^2 \quad \forall \begin{array}{l} q = 1, \ldots, N, \\ j = 1, \ldots, J-1. \end{array} \tag{B-11}$$

(v) The identities in (B-10) hold and the filters in the filter bank $\eta$ satisfy

$$\left|\widehat{b^{(0)}}\left(\frac{\delta_q}{2^j}\right)\right|^2 + \sum_{r=1}^k \left|\widehat{b^{(r)}}\left(\frac{\delta_q}{2^j}\right)\right|^2 = 1 \quad \forall q \in \theta_\delta^{(j)}, \ j = 2, \ldots, J, \tag{B-12}$$

with

$$\theta_\delta^{(j)} := \left\{ q \in \{1, \ldots, N\} : \widehat{\varphi}\left(\frac{\delta_q}{2^j}\right) \neq 0 \right\}.$$

**Proofs.** (i)$\Longleftrightarrow$(ii). Let $\Gamma_j^{(\text{low})} := \text{span}\{\Phi_{j,v} : v \in \mathcal{V}\}$ and $\Gamma_j^{(\text{high,r})} := \text{span}\{\Psi_{j,v}^{(r)} : v \in \mathcal{V}\}$. Define projections $\mathbf{P}\Gamma_j^{(\text{low})}, \mathbf{P}\Gamma_j^{(\text{high,r})}$ (with $r = 1, \ldots, k$) by

$$\mathbf{P}\Gamma_j^{(\text{low})}(f) := \sum_{v \in \mathcal{V}} \langle f, \Phi_{j,v} \rangle \Phi_{j,v}, \quad \mathbf{P}\Gamma_j^{(\text{high,r})}(f) := \sum_{c \in \mathcal{V}} \left\langle f, \Psi_{j,v}^{(r)} \right\rangle \Psi_{j,v}^{(r)}, \quad f \in l_2(\mathcal{G}_s). \tag{B-13}$$

Since $\mathbf{TiFraS}_{J_1}^J(\mathcal{S}, \eta; \mathcal{G}_s)$ is a tight frame on signed hypergraphs for $l_2(\mathcal{G}_s)$ ($J_1 = 1, \ldots, J$), we obtain by polarization identity,

$$f = \mathbf{P}\Gamma_{J_1}^{(\text{low})}(f) + \sum_{j=J_1}^J \sum_{r=1}^k \mathbf{P}\Gamma_j^{(\text{high,r})}(f) = \mathbf{P}\Gamma_{J_1+1}^{(\text{low})}(f) + \sum_{j=J_1+1}^J \sum_{r=1}^k \mathbf{P}\Gamma_j^{(\text{high,r})}(f) \tag{B-14}$$

for all $f \in l_2(\mathcal{G}_s)$ and for all $J_1 = 1, \ldots, J$. Thus, for $J_1 = 1, \ldots, J-1$,

$$\mathbf{P}\Gamma_{J_1+1}^{(\text{low})}(f) = \mathbf{P}\Gamma_{J_1}^{(\text{low})}(f) + \sum_{r=1}^k \mathbf{P}\Gamma_{J_1}^{(\text{high,r})}(f), \tag{B-15}$$

which is (B-7).

Moreover, when $J_1 = J$, (B-14) gives (B-6). Consequently, (i)$\Longrightarrow$(ii). Conversely, recursively using (B-15) gives

$$\mathbf{P}\Gamma_{m+1}^{(\text{low})}(f) = \mathbf{P}\Gamma_{J_1}^{(\text{low})}(f) + \sum_{j=J_1}^m \sum_{r=1}^k \mathbf{P}\Gamma_{J_1}^{(\text{high,r})}(f) \tag{B-16}$$

for all $J_1 \leq m \leq J-1$. Taking $m = J-1$ together with (B-6), we deduce (B-14), which is equivalent to (B-5). Thus, (ii)$\Longrightarrow$(i).

(ii)$\Longleftrightarrow$(iii). The equivalence between (ii) and (iii) simply follows from the polarization identity.

(ii)$\Longleftrightarrow$(iv). By the orthonormality of $\boldsymbol{m}_p$,

$$\langle f, \Phi_{j,v} \rangle = \sum_{v=1}^N \widehat{\varphi}\left(\frac{\delta_q}{2^j}\right) \widehat{f}_q\, m_q(v), \quad \left\langle f, \Psi_{j,q}^{(r)} \right\rangle = \sum_{q=1}^N \widehat{\psi^{(r)}}\left(\frac{\delta_q}{2^j}\right) \widehat{f}_q\, m_q(v),$$

where $\widehat{f}_q = \langle f, \boldsymbol{m}_q \rangle$ is the Fourier coefficient of $f$ with respect to $\boldsymbol{m}_q$. This together with (B-13), (B-2) and (B-3) gives, for $j \geq 1$ and $r = 1, \ldots, k$, the Fourier coefficients for the projections $\mathbf{P}\Gamma_j^{(\text{low})}(f)$ and $\mathbf{P}\Gamma_j^{(\text{high,r})}(f)$:

$$\left(\widehat{\mathbf{P}\Gamma_j^{(\text{low})}(f)}\right)_q = \left|\widehat{\varphi}\left(\frac{\delta_q}{2^j}\right)\right|^2 \widehat{f}_q, \quad \left(\widehat{\mathbf{P}\Gamma_j^{(\text{high,r})}(f)}\right)_q = \left|\widehat{\psi^{(r)}}\left(\frac{\delta_p}{2^j}\right)\right|^2 \widehat{f}_q, \quad \forall q = 1, \ldots, N, \tag{B-17}$$

which implies that (B-6) and (B-7) are equivalent to (B-10) and (B-11) respectively. Thus, (ii)$\Longleftrightarrow$(iv).

(iv)$\Longleftrightarrow$(v). Based on the relations (B-1) that $\widehat{\varphi}(2\xi) = \widehat{b^0}(\xi)\widehat{\varphi}(\xi)$ and $\widehat{\psi^{(r)}}(2\xi) = \widehat{b^{(r)}}(\xi)\widehat{\varphi}(\xi)$ for any $\xi \in \mathbb{R}$, it can be deduced that for $q = 1, \ldots, N$ and $j \geq 1$,

$$\left|\widehat{\varphi}\left(\frac{\delta_q}{2^j}\right)\right|^2 + \sum_{r=1}^{k}\left|\widehat{\psi^{(r)}}\left(\frac{\delta_q}{2^j}\right)\right|^2 = \left(\left|\left(\frac{\delta_q}{2^{j+1}}\right)\right|^2 + \sum_{r=1}^{k}\left|\left(\frac{\delta_q}{2^{j+1}}\right)\right|^2\right)\left|\widehat{\varphi}\left(\frac{\delta_q}{2^{j+1}}\right)\right|^2.$$

This shows that (B-11) is equivalent to (B-12). Therefore, (iv)$\Longleftrightarrow$(v).

## C. More Details on Datasets

In our experimental studies, we utilize five real-world datasets derived from courses at three universities: the *Biology* and *Law* courses from the University of Auckland, the *Cardiff20102* Medical School course from Cardiff University, and the biochemistry courses *Sydney19351* and *Sydney23146* from the University of Sydney. These datasets serve as the foundation for the student performance prediction task. To align with the problem formulation, we construct signed hypergraphs based on student-question interaction data. This appendix provides further details on the process of defining signed (bipartite) hypergraphs for each dataset, which serves as the input signal for the problem-solving framework (as illustrated in Figure 1). Additionally, to demonstrate the advantages of using signed (bipartite) hypergraphs over signed (bipartite) graphs, we also present the construction of signed graphs, which allows for comparative experimental studies and provides further insights into the merits of EduLLM (as detailed in the following Appendix E.3).

**Formulating Original '*Students-MCQs*' Sources as Signed Hypergraph.** To capture the complex interactions between students and questions, we constructed signed hypergraphs using students' answer records. Signed hypergraphs are particularly effective in modeling higher-order relationships, as they allow us to categorize student responses into "correct" or "incorrect" answers, with these responses forming hyperedges in the data. Specifically:

- For each question node $q_j$, all students who correctly answered the question are connected to the question node via a positive hyperedge, which is labeled as "+1".

- In contrast, students who answered the question incorrectly are connected to the question node through a negative hyperedge, labeled as "-1".

This structure enables the representation of each dataset as a signed hypergraph, providing a more expressive framework for modeling student-question interactions. By capturing higher-order relationships, signed hypergraphs offer a richer representation compared to traditional signed graphs, overcoming their limitations. For a better clarity, a toy example for signed hypergrpah construction is provided in the following **Appendix D**.

**Formulating Original '*Students-MCQs*' Sources as Signed Graph (as a Specific Comparison Case).** In comparison, signed graphs represent a simpler structure by focusing on individual interactions between students and questions. For each dataset, we constructed a signed bipartite graph using the answer records:

- If a student $u_i$ answered a question $q_j$ correctly, an edge with a positive sign ("+1") is established between the two nodes.

- If the student's answer was incorrect, an edge with a negative sign ("-1") is created between the student node and the question node.

The bipartite structure of signed graphs is more straightforward and is particularly suited for analyzing one-to-one relationships between students and questions. From our statistical analysis, we observe that the overall correctness rates for most courses ranged between 60% and 70%, except for the Law course, which exhibited an impressive correctness rate of 93%.

## D. Additional Details on EduLLM's Key Components

As outlined in Section 3.1, prior to the implementation of FraS-HNN, the core module for signed hypergraph learning, it is crucial to first establish the modules for LLM-based semantic extraction and signed hypergraph construction. This part provides further details on these two essential components.

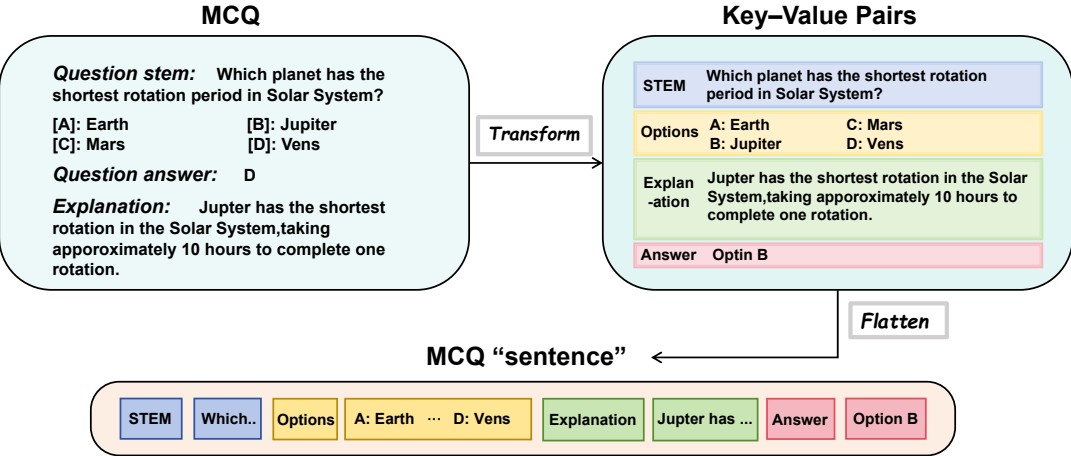

*Figure D-1.* Flatten MCQ component key-value pairs into a sequence form.

### D.1. LLM-based Semantic Extraction

We recognize the significant potential of integrating semantic context into multiple-choice questions (MCQs), as it can reveal latent knowledge that is crucial for prediction tasks. To enhance the representation of MCQs, we employ LLMs to integrate semantic information, which enables the fusion of textual features with signed hypergraph data. We also note that the procedure for LLM-based semantic extraction of questions is adapted from the related work by (Wang et al., 2024b). We revisit the general process below not only for clarity but also to help readers better understand how to apply different LLM tools or platforms in similar contexts in their future work.

MCQs are a cornerstone of educational assessments and are widely used in both traditional education and crowdsourced learning platforms. However, conventional methods for representing MCQs often fail to leverage the full potential of their semantic content, thereby limiting their utility in predictive modeling tasks. By incorporating semantic context, the representation of MCQs can be significantly enriched, leading to improved performance in downstream prediction tasks. Despite the broad application of MCQs in various educational contexts, the integration of textual and graph data to enhance MCQ representations has been underexplored in the literature, underscoring the need for novel approaches to address this gap.

In our EduLLM framework, we use LLMs to enrich MCQ representations with semantic context. The process begins by transforming the various components of an MCQ into a flattened sentence format, as shown in Figure D-1. Each question is reformulated into key-value pairs extracted from the question stem, answer options, and explanations. Using the LLM, we extract a set of knowledge point terms from each MCQ, denoted as $k_i$, and assign a corresponding weight to each term, denoted as $h_i$. Next, we compute word vectors for each term in the knowledge point set using GloVe embeddings (Pennington et al., 2014). The word vectors are then averaged to obtain a representation for each knowledge point, denoted as $kp_i$. The semantic representation of the entire MCQ is constructed by taking a weighted average of the knowledge points' vectors. The resulting semantic embedding for the question is denoted as $w_j$, which is computed as: $kp_i = (\sum_{t=1}^{m} f_{i,t})/m$, $w_j = (\sum_{i=1}^{n} h_i kp_i)/(\sum_{i=1}^{n} h_i)$. Here, $f_{i,t}$ denotes represents the embedding of the $t$-th term in the $i$-th knowledge point. These semantic embeddings are then combined with structural information from the graph data, resulting in a richer representation for each node within the graph.

For the semantic extraction process, we utilize ChatGPT as the LLM, which allows us to extract $n$ pairs of knowledge points ($k_i$ and $h_i$) in a single API request. Recall the schematic of EduLLM framwork shown in Figure 1, we design our prompts to efficiently extract relevant keywords and their associated weights, with results returned in a structured JSON format. This setup ensures that we can effectively handle large volumes of requests while maintaining the quality of the responses. The LLM outputs contain essential elements for understanding the MCQs, enabling it to identify specific knowledge areas related to the questions and generate meaningful semantic embeddings. A detailed description of the LLM API requests we have designed is provided in the following illustration.

- **Task Objective**: Your task is to extract key knowledge points from an MCQ created by a first-year student who studies Law in college. The question consists of a stem, up to five options, an answer, and an explanation provided by the student author.

- **Special Requirements**: Please provide your response in JSON format with the following keys and format:

  ```
  {"Keywords": [{"keyword": "keyword_name", "percentage": 0}]]}
  ```

  1. Each `"keyword_name"` should be a word or a short term with less than five words.
  2. The percentages of all keywords should add up to 1.
  3. Only include the top five keywords to avoid excessive keyword extraction.

- **In Context Learning**: Here's an example response format:

  ```
  {"Keywords": [
    {"keyword": "Legal person", "percentage": 0.8},
    {"keyword": "Intellectual property rights", "percentage": 0.2 }
  ]}
  ```

- **Context**: Here's the MCQ:

  Question stem: Which of the following situations is most likely to be considered a breach of contract?

  [A]: A party realizes after signing the contract that they are in financial difficulty and fails to make timely payments.

  [B]: The contract specifies the delivery of a particular product, but the delivering party provides a different substitute product.

  [C]: A party fails to fulfill their obligations on time due to force majeure (e.g., a natural disaster).

  [D]: Both parties mutually agree to terminate the contract during the performance phase.

  Question answer: B.

  Explanation: Option B best fits the definition of a breach of contract. A breach occurs when one party fails to perform its obligations according to the terms of the contract, or performs in a manner inconsistent with the contract's requirements. In this case, providing a substitute product instead of the one specifically agreed upon constitutes a violation of the contract terms. Option A: While failing to make a timely payment might lead to a breach, "financial difficulty" may be subject to further analysis and is not automatically considered a breach in every case. Option C: A failure to perform due to force majeure is generally regarded as an excusable non-performance, not a breach of contract. Option D: Mutually agreed termination of the contract is a legitimate contractual decision and does not constitute a breach.

- **LLM Response**:

  ```
  {
    "Keywords": [
      {"keyword": "breach of contract", "percentage": 0.6},
      {"keyword": "substitute product", "percentage": 0.2},
      {"keyword": "contract terms", "percentage": 0.1},
      {"keyword": "force majeure", "percentage": 0.05},
      {"keyword": "mutually agreed termination", "percentage": 0.05}
    ]
  }
  ```

### D.2. Toy Example for Signed Hypergraph Construction

Before proceeding to the signed hypergraph learning module, FraS-HNN, we first need to explicitly construct the structure of the signed hypergraph, as outlined in **Appendix C**. This is essential for capturing the higher-order relationships between students and MCQs. Specifically, to construct a signed hypergraph $\mathcal{G} = (\mathcal{V}, \mathcal{E})$, we need to define the vertex set $\mathcal{V}$ and the hyperedge set $\mathcal{E}$, along with the incidence matrix $H$, which reflects the underlying structure of the hypergraph:

**Vertex Set $\mathcal{V}$**: This set includes the set of students $\mathcal{U}$ and the set of questions $\mathcal{Q}$. Specifically, $\mathcal{U} = \{\text{Student 1}, \text{Student 2}, \ldots\}$ represents the set of students, and $\mathcal{Q} = \{\text{Question 1}, \text{Question 2}, \ldots\}$ represents the set of questions. There is no overlap between students and questions, i.e., $\mathcal{U} \cap \mathcal{Q} = \emptyset$.

**Hyperedge Set $\mathcal{E}$**: The hyperedge set is used to represent interactions between students and questions. Based on the answers provided by students, interactions are classified into positive and negative hyperedges. Specifically, positive hyperedges

connect all students who answered a question correctly to the corresponding question, while negative hyperedges connect all students who answered incorrectly to the corresponding question. The positive hyperedge set is denoted as $\mathcal{E}^+$, and the negative hyperedge set as $\mathcal{E}^-$. Each positive hyperedge $e^+ \in \mathcal{E}^+$ connects all students who answered correctly, and each negative hyperedge $e^- \in \mathcal{E}^-$ connects all students who answered incorrectly.

**Construction of the Incidence Matrix.** To facilitate further processing and computation, we use an incidence matrix $H \in \{0, 1, -1\}^{N \times M}$ to represent interactions between students and questions. Here, $N$ is the number of vertices (the sum of the number of students and the number of questions), and $M$ is the number of hyperedges (the total number of positive and negative hyperedges). The matrix construction is as follows:

- $H(v, e^+) = 1$, if $v$ belongs to a positive hyperedge $e^+$;

- $H(v, e^-) = -1$, if $v$ belongs to a negative hyperedge $e^-$;

- $H(v, e) = 0$ otherwise.

For example, consider a scenario with 3 students and 2 questions. The interaction data between students and questions is presented in Table D-1. In this table, a checkmark ($\checkmark$) represents a correct answer, and a cross ($\times$) represents an incorrect answer.

*Table D-1.* A simple example for three students and two questions.

| Student ID | Question 1 | Question 2 |
|---|---|---|
| student 1 | $\checkmark$ | $\times$ |
| student 2 | $\times$ | $\checkmark$ |
| student 3 | $\checkmark$ | $\checkmark$ |

Based on the above interaction data, the incidence matrix $H$ is constructed as follows:

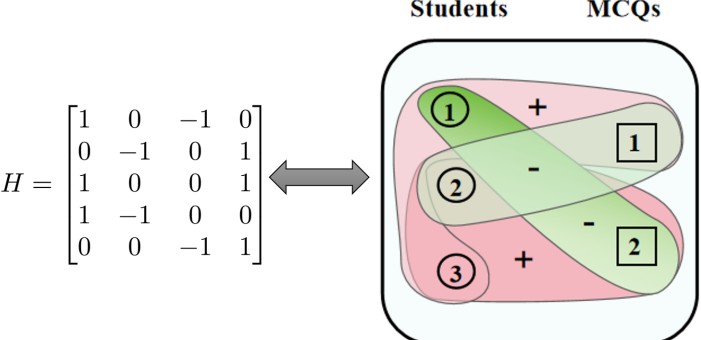

$$H = \begin{bmatrix} 1 & 0 & -1 & 0 \\ 0 & -1 & 0 & 1 \\ 1 & 0 & 0 & 1 \\ 1 & -1 & 0 & 0 \\ 0 & 0 & -1 & 1 \end{bmatrix}$$

*Figure D-2.* Incidence matrix and illustration of the signed hypergraph.

Clearly, this signed hypergraph consists of five nodes and four hyperedges: $e_1^+$ (Student 1, Student 3, Question 1), $e_2^+$ (Student 2, Student 3, Question 2), $e_1^-$ (Student 1, Question 2), and $e_2^-$ (Student 2, Question 1).

# E. Additional Experimental Details and Further Results

### E.1. Details for Baseline Models

In our experiments, we use several baseline models for performance comparison, including Random Embedding, Graph Convolutional Network (GCN) (Kipf & Welling, 2017), Graph Attention Network (GAT) (Veličković et al., 2018), Signed Graph Convolutional Network (SGCN) (Derr et al., 2018), Signed Bipartite Graph Neural Network (SBGNN) (Huang et al., 2021), Signed Bipartite Graph Contrastive Learning (SBCL) (Wang et al., 2024b), and its variant version LLM-SBCL (Wang et al., 2024b). Below is a brief description of each model:

- **Random Embedding:** This method generates random 64-dimensional embeddings for both students ($z_{u_i} \in \mathbb{R}^{64}$) and questions ($z_{v_j} \in \mathbb{R}^{64}$). These embeddings are concatenated, and a Logistic Regressor (LR) is trained on the

concatenated embeddings to predict the sign in the test data. Since random embeddings do not incorporate any graph-related information, this model serves as a lower-bound benchmark (Huang et al., 2021).

- **GCN (Kipf & Welling, 2017):** It utilizes message-passing based graph convolution to aggregate features from nodes and their neighbors, capturing local graph structure information. Its hierarchical design, which allows stacking multiple convolutional layers, enables the extraction of deeper graph features, making GCN suitable for node classification and graph representation learning tasks.

- **GAT (Veličković et al., 2018):** It introduces a self-attention mechanism that assigns different weights to each node, enabling the model to adapt to diverse graph structures for more flexible feature aggregation. This dynamic weighting mechanism allows GAT to explicitly assess and quantify the importance of neighboring nodes, resulting in more precise representation learning.

- **SGCN (Derr et al., 2018):** It leverages balance theory to incorporate negative links during aggregation. It addresses the unique challenges posed by negative links, which carry distinct semantic meanings and form complex relationships with positive links, enabling more accurate modeling of signed graphs.

- **SBGNN (Huang et al., 2021):** It employs advanced techniques for modeling signed bipartite graphs. We use the publicly available code [1] with default settings and 300 training epochs for our implementation.

- **SBCL (Wang et al., 2024b):** It employs a single-headed GAT with two convolutional layers and 64-dimensional random embeddings. We use publicly available code [2] with default settings and 300 training epochs for this model.

- **LLM-SBCL (Wang et al., 2024b):** It is a variant of SBCL that integrates semantic embeddings for question components, which are generated by a large language model. This addition enhances the model's ability to better capture the semantic context of the questions.

### E.2. Experimental Setup and Hyperparameter Settings

All experiments are conducted using the PyTorch framework and executed on a single NVIDIA RTX A6000 GPU to ensure computational efficiency and consistency across all runs. To facilitate the reproducibility of our results, we provide the corresponding hyperparameters that yield optimal performance in these experiments. These hyperparameter settings are detailed in Table E-1, which allows for the accurate reproduction of the experimental setup and comparison of performance results.

### E.3. Further Study: EduLLM vs. EduLLM-SG (EduLLM with Signed GNNs)

The previous performance comparison between EduLLM and other baseline models has already demonstrated the superiority of EduLLM. This advantage stems from the new problem formulation using signed hypergraphs and the incorporation of key techniques, including both LLM-based semantic extraction and signed hypergraph learning. To further explore the benefits of signed hypergraphs over signed graphs in solving this problem, a natural question arises: "Assuming the influence of LLM is excluded, how does EduLLM perform when using signed graph learning while still employing framelet-based convolutions?" To answer this, we revisit the problem formulation by replacing the signed hypergraph with a signed graph (as described in **Appendix C**). In this case, EduLLM is refined by constructing framelet-based convolutions on the signed graph, resulting in a variant model, which we term EduLLM-SG, as illustrated in Figure E-1.

Specifically, the signed (bipartite) graph consists of two types of nodes: students and questions. Interactions between students and questions are represented by edges, where the sign of the edge indicates whether the student answered the question correctly (positive) or incorrectly (negative). This graph is denoted as $\mathcal{G} = (\mathcal{U}, \mathcal{V}, E)$, where $\mathcal{U} = \{u_1, u_2, \ldots, u_{|\mathcal{U}|}\}$ represents the set of students, $\mathcal{V} = \{v_1, v_2, \ldots, v_{|\mathcal{V}|}\}$ represents the set of questions, and there is no overlap between the two sets (i.e., $\mathcal{U} \cap \mathcal{V} = \emptyset$). The edge set $E \subset \mathcal{U} \times \mathcal{V}$ represents the pair-wise relationships between students and questions, divided into positive edges ($E^+$) and negative edges ($E^-$), with $E = E^+ \cup E^-$, and $E^+ \cap E^- = \emptyset$.

For a fair comparison, we apply the same experimental setups and hyperparameter settings, as detailed in Table E-1, to the experimental study on EduLLM-SG. The results for both EduLLM-SG and EduLLM are presented in Table E-2. The

---

[1]https://github.com/huangjunjie-cs/SBGNN
[2]https://github.com/Alex-Zeyu/SBGCL

*Table E-1.* Hyperparameter settings for the five datasets used in the experiments.

| Dataset | Hyperparameter Setting | |
|---------|---------|---------|
| **Biology** | Learning rate: 5e-3
Weight decay: 1e-4
Hidden Size: 64
Dropout ratio: 0.2
Level: 1 | Layers: 36
Alpha: 0.5
Gamma: 0.5
Lambda: 0.8
Seed: 2000 |
| **Law** | Learning rate: 5e-3
Weight decay: 1e-4
Hidden Size: 64
Dropout ratio: 0.2
Level: 2 | Layers: 16
Alpha: 0.5
Gamma: 0.5
Lambda: 0.8
Seed: 2000 |
| **Cardiff** | Learning rate: 5e-2
Weight decay: 1e-2
Hidden Size: 64
Dropout ratio: 0.2
Level: 2 | Layers: 24
Alpha: 0.1
Gamma: 0.1
Lambda: 0.5
Seed: 2000 |
| **Sydney19** | Learning rate: 1e-2
Weight decay: 1e-3
Hidden Size: 64
Dropout ratio: 0.2
Level: 2 | Layers: 28
Alpha: 0.2
Gamma: 0.2
Lambda: 0.6
Seed: 2000 |
| **Sydney23** | Learning rate: 5e-2
Weight decay: 1e-4
Hidden Size: 64
Dropout ratio: 0.2
Level: 2 | Layers: 16
Alpha: 0.1
Gamma: 0.1
Lambda: 0.5
Seed: 2000 |

*Table E-2.* Comparison of EduLLM and EduLLM-SG Performance

| | EduLLM-SG | EduLLM |
|---|---|---|
| Biology | $0.807 \pm 0.007$ | $\mathbf{0.809 \pm 0.010}$ |
| Law | $0.941 \pm 0.030$ | $\mathbf{0.945 \pm 0.005}$ |
| Cardiff | $0.748 \pm 0.003$ | $\mathbf{0.753 \pm 0.011}$ |
| Sydney19 | $\mathbf{0.715 \pm 0.012}$ | $0.712 \pm 0.016$ |
| Sydney23 | $0.823 \pm 0.014$ | $\mathbf{0.829 \pm 0.006}$ |

experimental results clearly show that EduLLM consistently outperforms EduLLM-SG, validating the effectiveness of using signed hypergraphs in this specific educational application. By capturing higher-order interactions, the signed hypergraph framework enables the model to more comprehensively represent the complex relationships between students and questions, offering a richer and more expressive model than the signed graph approach. This advantage is especially evident in datasets such as Biology and Law. However, for the Sydney19 dataset, EduLLM-SG achieves slightly better performance than EduLLM. We hypothesize that this small difference may be due to the limited number of answer records and high data sparsity in the dataset. In such sparse scenarios, the higher-order relationships captured by hypergraphs may not provide enough discriminative power and may even introduce redundant information, leading to reduced performance. In contrast, the simplicity of signed graphs can offer more effectiveness in these conditions.

Interestingly, despite not outperforming EduLLM, EduLLM-SG still surpasses all other baseline models recorded in Table 2 in the main manuscript, especially those that also use signed graphs (such as SGCN (Derr et al., 2018), SBGNN (Huang et al., 2021), SBCL (Wang et al., 2024b), and LLM-SBCL (Wang et al., 2024b)). This further highlights the benefits of the FraS-GNN approach (with the advantages of low-pass and high-pass filters inherited from FraS-HNN, as theoretically studied in **Appendix B**) in signed graph learning. Overall, the experimental results presented in Table E-2, along with the previous findings, convincingly demonstrate the effectiveness and advantages of EduLLM in this task.

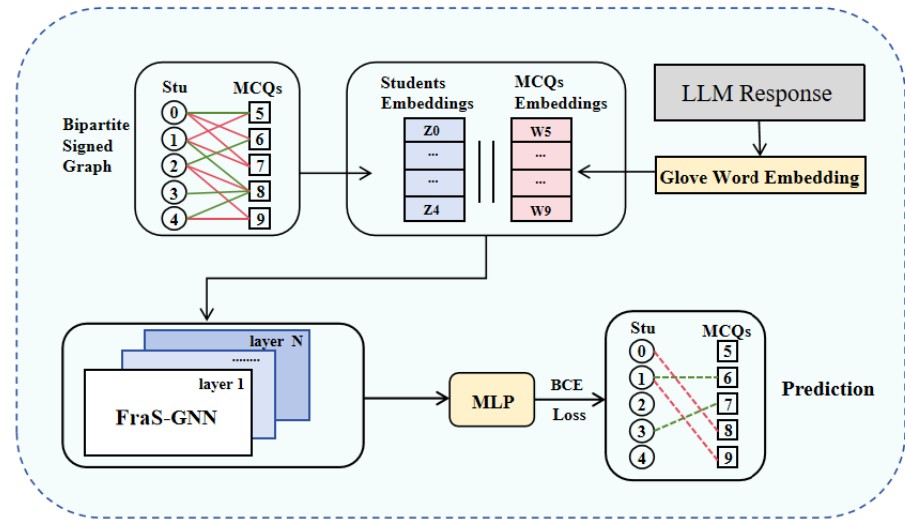

*Figure E-1.* The framework of EduLLM-SG

## E.4. Additional Results on Parameter Sensitivity Analysis for EduLLM and EduLLM-SG

In line with the results presented in Figure 2 of the main manuscript, we provide additional empirical results on parameter sensitivity analysis for EduLLM in Figure E-2 and for EduLLM-SG in Figure E-3. Overall, both models demonstrate low sensitivity to variations in the parameters $\alpha$, $\gamma$, and $\lambda$ across most cases.

# F. Further Highlights

We emphasize that although EduLLM in its current form is specifically designed for student performance prediction, an important problem in educational data analytics and mining, the underlying signed hypergraph neural network (FraS-HNN) has broader applicability. It holds strong potential for tackling analogous tasks in other domains and for advancing the development of hypergraph-based machine learning models and techniques (Li et al., 2025a;b). This is particularly true when signed hypergraph learning can offer effective solutions to the problem at hand. Signed hypergraphs are particularly effective for problems involving multi-party interactions, where both positive and negative relationships need to be modeled, making them suitable for various real-world applications. Thus, the potential for applying FraS-HNN or its future variants goes beyond educational contexts, offering promising avenues for research and application across various interdisciplinary fields.

For future work, we foresee the application of signed hypergraph learning (SHL) to a variety of specific tasks across different fields, provided that the data sources and problem formulations align well with the SHL paradigm. Furthermore, to advance research in SHL, the development of benchmark datasets of signed hypergraphs across various domains is highly expected.

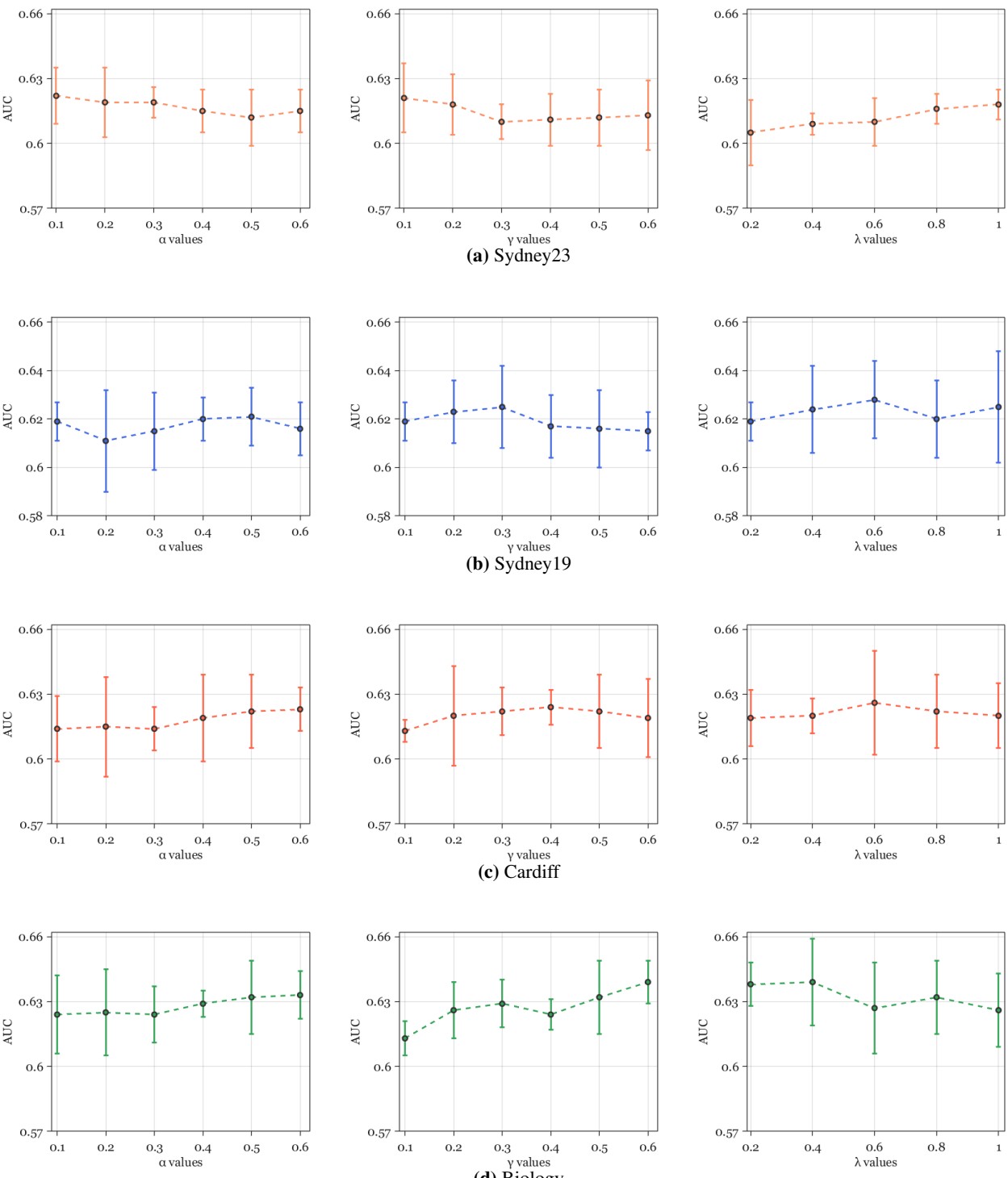

*Figure E-2.* Parameter sensitivity analysis for EduLLM across the Sydney23, Sydney19, Cardiff, and Biology datasets.

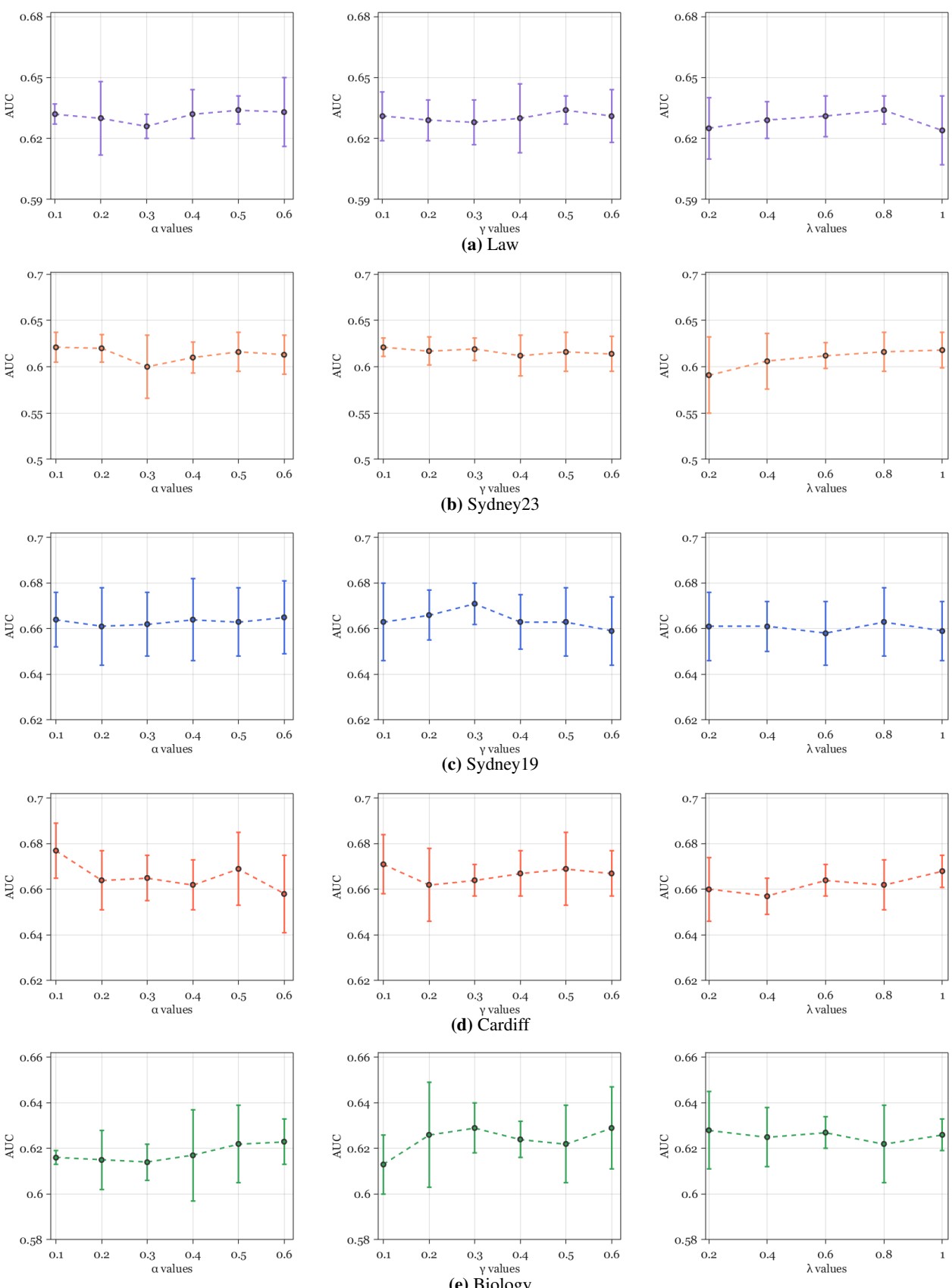

*Figure E-3.* Parameter sensitivity analysis for EduLLM-SG across the Law, Sydney23, Sydney19, Cardiff, and Biology datasets.

