# OpenReview forum: "EduLLM: Leveraging Large Language Models and Framelet-Based Signed Hypergraph Neural Networks for Student Performance Prediction"
_ICML.cc/2025/Conference — ICML 2025 poster_

### Official Review · Reviewer_vEwM · 2025-03-09

**Overall Recommendation:** 3

**Summary:**

The paper introduces EduLLM, a new framework for student performance prediction that combines Large Language Models (LLMs) with a Framelet-based Signed Hypergraph Neural Network (FraS-HNN). FraS-HNN is a novel approach for signed hypergraph learning, utilizing high-pass and low-pass filters to extract multi-frequency interactions between students and questions. LLMs are used to enhance semantic representation, complementing hypergraph structural learning to improve predictive accuracy. The LLM-based semantic feature extraction is relatively standard, with limited discussion on LLM selection, fine-tuning, or task adaptation. It appears to mainly convert text into embeddings and integrate them with the hypergraph structure.

**Claims And Evidence:**

Claims
1. EduLLM improves student performance prediction by integrating LLMs with hypergraph neural networks.
2. FraS-HNN effectively models signed hypergraphs, capturing both structural and semantic information to enhance prediction performance.
3. EduLLM outperforms existing state-of-the-art (SOTA) methods across multiple educational datasets.
Claims 1 and 2 are supported by experiments but lack a detailed discussion on optimizing the LLM component. LLMs are primarily used for feature extraction without further fine-tuning or analysis.
Claim 3 shows strong results on five datasets, but th comparison lacks larger-scale datasets and does not evaluate computational complexity.
The theoretical analysis of FraS-HNN is complex but lacks an intuitive explanation of its advantages over existing methods.

**Essential References Not Discussed:**

1. HyperGCN (Yadati et al., 2019): A strong baseline for hypergraph convolution, should be included in comparative experiments.
2. AKT (Ghosh et al., 2020): A state-of-the-art knowledge tracing model, relevant for student performance prediction, should be compared against EduLLM.
3. MOOC-related work (Piech et al., 2015; Nakagawa et al., 2019): Prior studies evaluating large-scale online learning datasets—EduLLM should be tested on similar datasets for practical validation.

**Experimental Designs Or Analyses:**

The model's performance is evaluated on five educational datasets, showing that EduLLM consistently achieves higher F1-score and AUC than SOTA methods.
Ablation studies analyze the contributions of high-pass filters, low-pass filters, and LLMs.

Limitations:
1. Lack of large-scale dataset evaluation—current datasets have limited diversity and scale.
2. No computational complexity analysis—FraS-HNN's scalability remains unclear.

**Methods And Evaluation Criteria:**

1. The paper models student-question interactions using signed hypergraphs and integrates LLM-enhanced semantic features, which provides a degree of novelty.
2. FraS-HNN is mathematically analyzed, including its framelet-based hypergraph filtering approach.

**Other Comments Or Suggestions:**

Discuss LLM selection and optimization—a comparison between different LLMs (e.g., GPT-4, LLaMA) would strengthen the contribution.

**Other Strengths And Weaknesses:**

Strengths:
1. EduLLM achieves SOTA performance on the evaluated datasets.
2. Integrating LLMs with hypergraph neural networks introduces a new perspective for student modeling.

Weaknesses:
1. The study only evaluates five small datasets, lacking tests on more challenging real-world datasets (e.g., MOOCs).
2. The paper does not discuss the scalability of FraS-HNN on large-scale datasets.

**Questions For Authors:**

How was the LLM chosen? Have different LLM architectures been considered?

**Relation To Broader Scientific Literature:**

1. EduLLM uses FraS-HNN for signed hypergraph learning, related to HyperGCN, HCHA, and UniGNN.
2. The work can be seen as an extension of KT tasks, related to models such as DKT, DKVMN, AKT, and SAINT.
3. The paper applies LLMs for semantic feature extraction, aligning with recent studies on ChatGPT in personalized learning.

**Theoretical Claims:**

Theoretical results demonstrate multi-frequency signal analysis and the design of high-pass/low-pass filters.However, comparative experiments do not show a clear advantage of FraS-HNN over traditional hypergraph GNNs (e.g., HyperGCN, HCHA).

---

> ### Author Rebuttal · Authors · 2025-03-28
>
> We sincerely thank the reviewer for the constructive feedback and for acknowledging the novelty and strengths of our proposed framework. Please find our detailed responses below:
> - **LLM Selection, Fine-tuning, and Task Adaptation:**  We appreciate the reviewer’s observation and agree that further analysis of LLM choices could be valuable in broader contexts. However, to clarify the scope of our work: LLMs in EduLLM are primarily used as a semantic information extraction tool to generate initial node features for questions in the student-question signed hypergraph, rather than being the core focus of technical innovation. **One reason** why we do not study in detail the effect of different LLM selections is that, as we would like to note, this submission is flagged under the **''Application-Driven Machine Learning''** category, where the focus is on solving a specific real-world task and not on advancing LLM development itself. **Another reason** is that we aimed to ensure fairness in the performance comparison with the baseline models, so we used the same LLM module and processing pipeline to generate the preprocessed semantic embeddings. This manner ensures that any observed improvements are due to the merits of the proposed framework, particularly the signed hypergraph setting and advantages of FraS-HNN, rather than differences in the LLM processing itself.
>
>   Alternatively, to approximately assess the impact of LLM-based embeddings on model performance, we conducted a robustness study presented in **Section 4.7**. The results, shown in **Figure 3**, demonstrate that the model’s performance degrades smoothly and only slightly at higher noise levels. This robustness suggests that EduLLM is relatively insensitive to variations in the LLM-induced embeddings, implying that replacing the current LLM backbone with a different one is unlikely to significantly affect task performance.
> - **Large-scale Dataset:**  We agree that evaluating on larger-scale datasets would strengthen the practical validation of EduLLM. Our current datasets are widely used and shared with baseline models, ensuring fairness and comparability. Meanwhile, our lab is constructing new large-scale datasets with MCQ texts from various subjects, which will allow us to further assess scalability and real-world applicability in future work.
> - **Theoretical Clarity and Intuition:**  While detailed proofs are provided in the appendix, we agree that offering more intuitive explanations would help clarify the motivation and benefits of our approach. In a future updated version, we will provide more intuitive explanations of FraS-HNN’s advantages over existing methods, which we believe will also benefit researchers working on **(signed) hypergraph learning**.
> - **Comparisons with Existing Hypergraph and KT Models:**  We would like to clarify that our problem formulation, predicting student performance at the question level using signed hypergraphs, is not entirely equivalent to the KT task, which (although relates to student performance prediction) typically focuses on modeling students' evolving mastery over concepts across time. We have added additional evaluations against representative hypergraph neural networks (HNNs), including **HGNN, HyperGCN, AllDeepSets, AllSetTransformer, ED-HNN, SheafHyperGNN**. Due to space constraints, the detailed results are provided in our response to **Reviewer stfm**, demonstrating clearly FraS-HNN's effectiveness in capturing **signed high-order interactions**.
> - **Computational Complexity and Scalability:**  We provide a comparative analysis of training complexity for several representative HNNs and our proposed FraS-HNN, summarized in the table below. In our formulation, $k$ (the number of high-pass filters), $J$ (the scale level in FraS-HNN), and $K$ (the largest number of non-zero values in the framelet transform matrices) are constants independent of specific hypergraph structures. In practice, $k$ and $J$ are small, and due to the sparsity of hypergraph framelets, $K$ is typically small and often comparable to $\|H\|_0$. As a result, FraS-HNN's overall complexity is comparable to models such as AllDeepSets and ED-HNN, without introducing significant overhead.
> | Model                | Computational Complexity |
> |---------------------|----------------------------------|
> | UniGCNII            | $\mathcal{O}(TL(N+M+\|H\|_0)d + TLNd^2)$ |
> | Deep-HGCN           | $\mathcal{O}(TLM'd + TLNd^2)$ |
> | AllDeepSets         | $\mathcal{O}(TL\|H\|_0d + TL(N+M)d^2)$ |
> | ED-HNN              | $\mathcal{O}(TL\|H\|_0d + TL(N+M)d^2)$ |
> | **FraS-HNN (ours)** | $\mathcal{O}(TL(kJ+1)Kd + TL(N+M)d^2)$ |
>
>   Here, $N$ and $M$ denote the number of nodes and hyperedges, respectively; $\|H\|_0$ is the number of non-zero entries in the incidence matrix; $T$, $L$, and $d$ denote the number of epochs, layers, and feature dimensions; $M^{'}$ is the number of edges in the clique expansion (when transforming the hypergraph into a graph).

---

### Official Review · Reviewer_xwCf · 2025-03-10

**Overall Recommendation:** 5

**Summary:**

This paper presents EduLLM, a novel framework that integrates Large Language Models (LLMs) with a Framelet-based Signed Hypergraph Neural Network (FraS-HNN) to address student performance prediction in personalized education systems. EduLLM models the complex structural and semantic relationships between students and multiple-choice questions (MCQs) by constructing signed hypergraphs, where positive and negative hyperedges capture correct and incorrect responses. LLMs provide fine-grained semantic embeddings for educational content, which are combined with the multi-frequency features extracted by FraS-HNN using low-pass and high-pass filters. Through comprehensive experiments on five real-world educational datasets, EduLLM demonstrates significant improvements over strong baselines, highlighting its effectiveness in both capturing high-order relationships and integrating semantic information for performance prediction tasks.

**Claims And Evidence:**

Yes, the claims made for EduLLM in this submission are well-supported by comprehensive evidence, including both rigorous theoretical analysis and extensive empirical results.

**Essential References Not Discussed:**

The paper cites a solid range of related works across knowledge tracing, signed graph learning, hypergraph neural networks, and educational data mining. The literature review is up to date.

**Experimental Designs Or Analyses:**

Yes, I have reviewed the experimental design and analyses. The experiments are well-structured, using five real-world educational datasets that are appropriate for the student performance prediction task. The evaluation includes comparisons against strong baselines, thorough ablation studies to assess the contributions of key components, parameter sensitivity analyses, and robustness tests for LLM-induced semantic representations.
Overall, the experimental designs and analyses are comprehensive.

**Methods And Evaluation Criteria:**

Yes, the proposed methods and evaluation criteria are well-aligned with the problem of student performance prediction.

**Other Comments Or Suggestions:**

See the above weaknesses

**Other Strengths And Weaknesses:**

S1: [New Problem Formulation] The paper introduces a well-defined and meaningful problem setting by modeling student performance prediction with signed hypergraphs to capture both correct and incorrect interactions in a high-order structure.
S2: [Novel Framework] The proposed EduLLM framework uniquely combines LLM-based semantic embeddings with a framelet-based signed hypergraph neural network, offering an innovative solution to educational prediction tasks.
S3: [Comprehensive Theoretical and Empirical Studies] The work provides both solid theoretical foundations and extensive empirical validation through detailed experiments, ablation studies and evaluation on multiple real-world datasets.

W1: The application and evaluation are focused on educational data, with limited discussion on how the framework could generalize to other domains or tasks beyond student performance prediction. Any further insights?
W2: Although I am familiar with framelets and wavelets, I am curious whether the current framework could be further enhanced by exploring alternative wavelet designs on hypergraphs. Specifically, would incorporating different types of wavelets lead to improved feature extraction or better adapt to various hypergraph structures?

**Questions For Authors:**

Q1: I am curious whether framelets can be designed specifically for directed and signed hypergraphs (I think this is a brand-new definition, to the best of my knowledge). Although this is beyond the scope of the current work, it would be interesting to understand the key challenges involved.

**Relation To Broader Scientific Literature:**

Technically, EduLLM advances the student performance prediction (which indeed is a classic educational data mining task) by incorporating LLM-generated embeddings to enrich the feature representations of questions, helping to model semantic nuances in student interactions.
Also, as for the problem-solving, prior studies have applied signed graphs to model positive and negative relationships (e.g., correct/incorrect answers) in educational scenarios. However, traditional signed graphs are limited to pairwise relations. EduLLM extends this by introducing signed hypergraphs, which can capture higher-order interactions (such as multiple students answering the same question) and better reflect the group dynamics present in educational settings.
EduLLM builds on this by applying framelet theory to signed hypergraphs, enabling the simultaneous extraction of global (low-pass) and discriminative (high-pass) features from complex educational interactions. I think this may potentially motivate more follow-up works on signed hypergraphs.

**Theoretical Claims:**

Yes, I have reviewed the provided theoretical analysis and proofs, particularly those related to the framelet-based signed hypergraph neural network (FraS-HNN). The mathematical formulations, including the construction of framelets, spectral filtering, and the tight frame properties, are clearly presented and appear logically sound.

---

> ### Author Rebuttal · Authors · 2025-03-26
>
> We sincerely thank the reviewer for the encouraging feedback on our work. We are pleased that you recognized **the novelty and technical contributions of EduLLM, including its theoretical soundness, methodological innovation, and comprehensive experimental validation**. For your key concerns, please find our responses below:
>
> - **Generalization Beyond Educational Domains & Alternative Wavelet Designs:**  Thank you for noting these issues. For detailed clarifications regarding the potential generalization of FraS-HNN beyond educational applications and the discussion on exploring alternative wavelet designs, please kindly refer to our responses to **Reviewer Lfh6** (who is also curious about this).  In addition, we would like to emphasize that, as flagged in our submission type, this work is submitted under the **Application-Driven ML Submissions** category. Our particular focus is specifically on the student performance prediction task, a well-recognized and meaningful problem in the educational domain. The proposed EduLLM framework is newly developed for solving this specific problem. Therefore, we did not engage in studying broader generalization across domains, as that falls outside the intended scope of our application-driven submission. For more context, please refer to the **Supplementary Guidelines for Reviewing Application-Driven ML Submissions** (see https://icml.cc/Conferences/2025/ReviewerInstructions).
>
>   Hopefully, this clarifies the position of our work within the specific context of this year’s ICML submission categories..
>
>
>
> - **Framelets for Directed and Signed Hypergraphs:**  We appreciate the reviewer’s insightful question regarding the extension of framelet theory to directed and signed hypergraphs. Although there is very recent work defining the notion of **directed hypergraphs** (see: [https://openreview.net/forum?id=h48Ri6pmvi](https://openreview.net/forum?id=h48Ri6pmvi)), to the best of our knowledge, a formal and unified definition of **directed and signed hypergraphs** has not yet appeared in the literature. Developing framelet transforms for such structures would require extending spectral theory to handle both edge directionality and polarity, potentially involving non-symmetric Laplacians or incidence-based spectral operators. Key challenges include defining appropriate inner product spaces, preserving desirable mathematical properties (e.g., tightness and localization), and ensuring the interpretability of the resulting representations. While this extension is beyond the scope of the current work, we view it as an important direction for future investigation. Of course, we hope that our current work, particularly the key module FraS-HNN, **can motivate further follow-up studies on directed and signed hypergraphs, whether through model development or advanced applications**.
>
> Again, we sincerely thank you for recognizing the merits of our proposed FraS-HNN module (also recognized by **Reviewer Lfh6**), which we believe contributes not only to the student performance prediction task **but also holds broader value for advancing (signed) hypergraph learning in general**. In response to **Reviewer stfm's** suggestions, we have conducted additional empirical studies that include comparisons with several representative hypergraph neural network (HNN) baselines, i.e. HGNN (AAAI, 2019), HyperGCN (NeurIPS, 2019), AllDeepSets (ICLR, 2022), AllSetTransformer  (ICLR, 2022), ED-HNN  (ICLR, 2023), and SheafHyperGNN (NeurIPS, 2023), by replacing the FraS-HNN backbone of EduLLM with each of these modules. Please refer to our response to Reviewer stfm for further details. These supplementary results further validate the effectiveness of the high-pass and low-pass filters in FraS-HNN, as positively noted in your comments. For future work, we plan to construct more challenging **signed hypergraph datasets beyond the educational domain (e.g., in social networks, biology, traffic systems)** to support the development of this emerging direction in terms of theoretical understanding, model design, and real-world applications. We believe our framework, along with the core FraS-HNN module, can benefit other complex tasks and applications where signed hypergraphs (or their variants) align well with the problem formulation.
>
>
> We hope the above responses have clarified your questions and comments. We welcome any further discussion or suggestions.

---

### Official Review · Reviewer_Lfh6 · 2025-03-10

**Overall Recommendation:** 5

**Summary:**

- This paper presents EduLLM, a novel method for predicting student performance by integrating LLM-based semantic understanding with structural modeling via a framelet-based signed hypergraph neural network (FraS-HNN).
- Signed hypergraphs capture higher-order interactions and differentiate correct from incorrect student responses, while LLMs enhance the semantic representation of questions.
- EduLLM leverages framelet transforms to extract both low- and high-frequency information from complex educational data.

**Claims And Evidence:**

- The claims in the submission are well-supported.

**Essential References Not Discussed:**

- This paper has appropriately cited relevant sources.

**Experimental Designs Or Analyses:**

- The use of diverse educational datasets, along with detailed comparisons and ablation analyses, provides solid evidence supporting the effectiveness.

**Methods And Evaluation Criteria:**

- EduLLM and evaluation criteria are well-suited for the student performance prediction,
- and benchmark datasets effectively show its effectiveness.

**Other Comments Or Suggestions:**

- To better highlight the advantages of signed hypergraph modeling, it would be helpful to include visual examples that compare the structural differences between traditional signed graphs and signed hypergraphs using real student-question interaction data.

**Other Strengths And Weaknesses:**

- Strengths
   - The paper redefines student performance prediction as a hypergraph learning, which effectively model both correct and incorrect student responses through positive and negative edges.
   - FraS-HNN applies multiscale signal processing with low-pass and high-pass filters to capture both shared patterns and individual differences within student interactions, enhancing the representation of complex educational relationships.
   - EduLLM successfully combines hypergraph structural learning with semantic features from LLMs, leading to a more comprehensive understanding of student-content interactions and delivering improved prediction performance across multiple datasets.
   - The theoretical analysis of FraS-HNN is sound.
- Weaknesses:
   - While the student performance prediction task is well-suited to the signed hypergraph learning formulation, a novel and insightful perspective proposed by the authors, EduLLM demonstrates strong performance on educational datasets.
      - However, the paper offers limited discussion on its potential generalization to other domains beyond student performance prediction.
   - The motivation for selecting the Haar-type filter in the framelet construction is not clearly explained.
      - The authors should elaborate on why this filter was chosen and whether other filter types were considered.
      - Additionally, could alternative filters offer advantages in capturing different patterns within the signed hypergraph?

**Questions For Authors:**

- The authors summarize the key insights of these properties in the main text.
- I strongly suggest that the authors enhance the clarity of their Abstract by clearly stating their motivation and key insights.

**Relation To Broader Scientific Literature:**

- The key contributions closely aligned with ongoing research in educational data mining, hypergraph learning, and the application of large language models.
- FraS-HNN introduces a novel method for hypergraph learning, which can be considered a model-level contribution, not confined to a specific application.
- FraS-HNN has the potential to inspire further advancements in hypergraph neural networks, hypergraph learning, and related applications.

**Theoretical Claims:**

- The theoretical proofs, formulations, and derivations are correct; however, the authors need to address some typos.

---

> ### Author Rebuttal · Authors · 2025-03-26
>
> We sincerely thank the reviewer for the thoughtful evaluation. We are pleased that the novelty and effectiveness of EduLLM and FraS-HNN are recognized, along with the **theoretical soundness, strong empirical results, and broader contributions to hypergraph learning and educational data mining**. Below we provide point-by-point clarifications regarding the concerns raised:
>
> - **Generalization Beyond Educational Domains:**   We agree that it is important to consider the applicability of FraS-HNN beyond the student performance prediction task. Although the current work is tailored for modeling educational data, **the proposed signed hypergraph formulation and framelet-based representation learning are inherently general and applicable to domains involving higher-order and polarity-sensitive interactions**. For instance, tasks such as signed social network modeling, sentiment-based recommendation, or misinformation spread detection can similarly benefit from the ability to distinguish positive and negative multi-way relations. The design of FraS-HNN does not rely on domain-specific assumptions, which supports its potential transferability.
>
> - **Motivation for Haar-type Filter:** Mathematically, the choice of the Haar-type filter is based on its efficiency, orthogonality, and suitability for decomposing signals into coarse (low-frequency) and detail (high-frequency) components. In the context of signed hypergraphs, this allows the model to simultaneously capture common learning behaviors and student-specific deviations. While other filters (such as Daubechies or spline-based filters) may offer smoother basis functions or better frequency localization, the Haar-type was selected as a starting point due to its simplicity and proven utility in prior work on multiscale graph signal processing. Exploring alternative filters remains a promising direction for future extension.
>
> - **Visual Comparison with Signed Graphs:**  We appreciate the suggestion to better highlight the advantages of signed hypergraph modeling. To support this, we will provide illustrative examples showing how signed hypergraphs can represent multi-way interactions (e.g., groups of students responding to related questions) with polarity, in contrast to signed graphs that are limited to pairwise relations.
>
> - **Further Insights and Clarification for the Theoretical Properties:** The intention behind our theoretical analysis is to reveal how **the proposed FraS-HNN effectively captures both low-pass and high-pass components of node signals in signed hypergraphs**, enabling the model to differentiate shared and individualized patterns within complex educational interactions. To make these insights more accessible to the readers, we will summarize the key takeaways in the main text (in the updated version), such as: (1) how the framelet transform enables multiscale analysis on signed hypergraphs, (2) the specific role of positive and negative hyperedges in modulating spectral responses, and (3) how this facilitates richer representation learning compared to traditional graph-based or unsigned hypergraph approaches.
>
> Again, we sincerely thank you for recognizing the merits of our proposed FraS-HNN module, which we believe contributes not only to the student performance prediction task **but also holds broader value for advancing (signed) hypergraph learning in general**. In response to **Reviewer stfm's** suggestions, we have conducted additional empirical studies that include comparisons with several representative hypergraph neural network (HNN) baselines, i.e., HGNN (AAAI, 2019), HyperGCN (NeurIPS, 2019), AllDeepSets (ICLR, 2022), AllSetTransformer  (ICLR, 2022), ED-HNN  (ICLR, 2023), and SheafHyperGNN (NeurIPS, 2023), by replacing the FraS-HNN backbone of EduLLM with each of these modules. Please refer to our response to Reviewer stfm for further details. These supplementary results further validate the effectiveness of the high-pass and low-pass filters in FraS-HNN, as positively noted in your comments. For future work, **we plan to construct more challenging signed hypergraph datasets beyond the educational domain (e.g., in social networks, biology, traffic systems)** to support the development of this emerging direction in terms of theoretical understanding, model design, and real-world applications. We believe our framework, along with the core FraS-HNN module, can benefit other complex tasks and applications where signed hypergraphs (or their variants) align well with the problem formulation.
>
> We hope the above responses clarify the key concerns raised in your comments and questions. Please feel free to reach out with any further suggestions or points for discussion. We are happy to engage further to improve the clarity and impact of our work.

---

### Official Review · Reviewer_stfm · 2025-03-12

**Overall Recommendation:** 2

**Summary:**

This paper introduces EduLLM, a framework that combines large language models (LLMs) with hypergraph learning to improve student performance prediction. Traditional methods mainly rely on historical response patterns but struggle to capture the complex interactions between students and learning content. To address this, EduLLM integrates FraS-HNN, a spectral-based model for signed hypergraph learning, where students and questions are represented as nodes, and response records are modeled as signed hyperedges to capture both structural and semantic relationships. FraS-HNN utilizes framelet-based filters to extract multi-frequency features, while EduLLM enhances predictions by incorporating fine-grained semantic features from LLMs. Experimental results on multiple datasets show that EduLLM outperforms existing approaches, demonstrating the effectiveness of combining LLMs with signed hypergraph learning.

**Claims And Evidence:**

Overall, the claims in this paper are fairly reasonable.

**Essential References Not Discussed:**

[1] Gao W, Liu Q, Huang Z, et al. RCD: Relation map driven cognitive diagnosis for intelligent education systems[C]//Proceedings of the 44th international ACM SIGIR conference on research and development in information retrieval. 2021: 501-510.

[2] Shao P, Yang Y, Gao C, et al. Exploring Heterogeneity and Uncertainty for Graph-based Cognitive Diagnosis Models in Intelligent Education[J]. arXiv preprint arXiv:2403.05559, 2024.

**Experimental Designs Or Analyses:**

I believe the experiments are quite insufficient. First, there are too few comparison methods. Why are only graph-based methods compared, and why are some common approaches in student performance prediction, such as cognitive diagnosis, not included? Additionally, since you are comparing graph-based methods, why not include some state-of-the-art graph representation learning methods, including hypergraphs and graph transformers?

**Methods And Evaluation Criteria:**

Overall, the proposed method appears relatively simple, which limits its level of innovation. Additionally, some design motivation lack sufficient explanation regarding their rationale and justification. For example:

1.	The introduction of LLMs in the method is solely for preprocessing raw text data—extracting keywords from multiple-choice question descriptions to construct a dictionary and using GloVe to learn representations as model inputs. However, I am curious why LLMs are considered an integral part of the model when they are merely used as a tool. There is no representation learning, fine-tuning, or specifically designed prompt engineering involved, and the approach relies on the simplest GloVe embeddings. Moreover, how is the stability of LLM outputs ensured during data processing, and how is their contribution controlled proportionally?

2.	The use of signed hypergraphs seems reasonable, but what is the specific novel design of this module in this paper? In particular, what is the motivation behind the framelet-based signed hypergraph convolution? How does this design specifically cater to the task of student performance prediction?

**Other Comments Or Suggestions:**

Refer to Strengths&Weaknesses.

**Other Strengths And Weaknesses:**

Overall, this paper needs improvement. At least in terms of its innovation and contribution to the specific field, it is limited, which is insufficient for ICML. Specifically, the paper has several notable weaknesses:

1.	The writing lacks logical clarity, and the motivation is not adequately addressed. Many sections are overly redundant and verbose.

2.	The innovation at the method level is limited, particularly in the special design and improvements related to student performance prediction that the authors emphasize. As I mentioned earlier.

3.	The experiments are insufficient and lack necessary state-of-the-art methods in the field.

4.	The descriptions and discussions of the experiments are limited, especially regarding the performance improvements of the model itself and the corresponding conclusions.

**Questions For Authors:**

Refer to Strengths&Weaknesses.

**Relation To Broader Scientific Literature:**

To be honest, the motivation of this paper is unclear. The authors state, "While effective, these methods often fall short in capturing the intricate interactions between students and learning content, as well as the subtle semantics of these interactions," which seems unreasonable. There has already been a substantial amount of research on cognitive diagnosis methods based on graph learning to explore the relationships between learners and learning elements. However, the authors do not mention or compare these existing approaches, which is quite surprising.

**Theoretical Claims:**

The theoretical section of this paper primarily discusses the properties of signed hypergraph neural networks, but it is unclear how these properties relate to the student performance prediction scenario being modeled.

---

> ### Author Rebuttal · Authors · 2025-03-28
>
> We sincerely thank the reviewer for carefully reading our paper and providing detailed feedback. We respectfully offer the following clarifications:
> - **Further Clarification on Motivation:** While cognitive diagnosis models have explored learner-element relationships using graphs, our goal is to provide a new perspective by modeling **student-question interactions using signed hypergraphs**. Specifically, in the context of **student performance prediction**, where responses can be either correct or incorrect, we introduce **positive and negative hyperedges** to explicitly encode this polarity in group-level interactions. We believe this formulation brings a fresh and principled structural view to the problem, which forms the core of our motivation.
> - **Clarifying the Role of LLMs:**  We would like to note that, this submission is flagged under the **Application-Driven Machine Learning** type, where the focus is on solving a specific real-world task rather than developing LLMs itself. In our framework, LLMs are used as a semantic feature extraction tool to generate question embeddings, serving as initial node features in the signed hypergraph. One reason we did not explore different LLM selection or prompt strategies is that we aimed to ensure fairness in performance comparison with the baseline models. We kept the use of the same LLM module and processing pipeline for generating the preprocessed semantic embeddings. This manner ensures that any observed improvements are due to the merits of the proposed framework, particularly the signed hypergraph setting and advantages of FraS-HNN, rather than differences in the LLM processing itself.
>
>   For your concern about **stability** and **proportional contribution** of LLM-induced embeddings, we actually conducted a robustness test in **Section 4.7**, where additive Gaussian noise was introduced to the semantic embeddings to simulate perturbations. The results (in Figure 3) indicate that performance degrades smoothly and moderately as the noise level increases, suggesting that EduLLM is generally **robust to fluctuations in LLM outputs**. To a certain extent, this empirical evidence supports the **stability** of the model with respect to semantic input variations. In addition, the **advantage of using LLM-induced semantic embeddings**, as opposed to initial embeddings, has already been validated in related works such as SBCL and LLM-SBCL.
> - **Comparisons and Baseline Selection:**  We would like to clarify that our problem formulation, i.e., modeling student-question interactions via **signed hypergraphs**, is fundamentally different from traditional **knowledge tracing** or **cognitive diagnosis** tasks. Specifically, cognitive diagnosis methods are designed around concept-level mastery modeling **over time** and typically require a fine-grained concept-question mapping. In contrast, our approach focuses on **question-level prediction** using **signed hyperedges** to represent correctness of responses, which is not directly supported by the data structures or assumptions of cognitive diagnosis datasets. Therefore, direct comparison would be misaligned and may lead to unfair conclusions. That said, we acknowledge the broader relevance of cognitive diagnosis research in educational modeling and will incorporate appropriate discussion in a future version. Additionally, based on your suggestion, we have included comparisons with hypergraph neural network (HNN) baselines, including **HGNN, HyperGCN, AllDeepSets, AllSetTransformer, ED-HNN, and SheafHyperGNN**, by replacing the FraS-HNN backbone of EduLLM with each of these HNN modules. As shown in the table below, EduLLM (with FraS-HNN) consistently outperforms the variants equipped with each HNN module across all datasets, demonstrating the effectiveness of FraS-HNN in modeling **signed high-order interactions** and its potential to benefit future research in **(signed) hypergraph learning**.
>
> |       Model\Dataset       |   Sydney19351   |   Sydney23146   |     Biology     |  Cardiff20102   |       Law       |
> |--------------------|----------------|-----------------|-----------------|-----------------|-----------------|
> |        HGNN          |   0.606±0.014   |   0.619±0.013   |   0.673±0.006   |   0.624±0.007   |   0.905±0.011   |
> |      HyperGCN        |   0.620±0.012   |   0.650±0.038   |   0.651±0.016   |   0.625±0.023   |   0.901±0.032   |
> |     AllDeepSets      |   0.626±0.017   |   0.660±0.012   |   0.697±0.008   |   0.637±0.023   |   0.898±0.009   |
> |  AllSetTransformer   |   0.618±0.022   |   0.661±0.010   |   0.689±0.016   |   0.644±0.029   |   0.906±0.009   |
> |       ED-HNN         |   0.662±0.023   |   0.708±0.032   |   0.715±0.039   |   0.673±0.026   |   0.910±0.011   |
> |   SheafHyperGNN      |   0.684±0.023   |   0.711±0.030   |   0.732±0.022   |   0.687±0.025   |   0.914±0.013   |
> | **EduLLM (with FraS-HNN)**| **0.712±0.016** | **0.829±0.006** | **0.809±0.010** | **0.753±0.011** | **0.945±0.005** |

---

> > ### Comment · Reviewer_stfm · 2025-04-06
> >
> > Thank you to the authors for their response, but I believe several of my concerns remain unaddressed:
> >
> > * The core contribution claimed by the authors is the use of signed hypergraphs to model student-question interactions from a new perspective. **However, this approach has already been explored in recent work on student modeling [1,2,3], where signed hypergraphs are essentially a combination of existing techniques**. Therefore, the claimed novelty of this work is rather limited.
> >
> > * The authors argue that the problem definition in this paper is fundamentally different from traditional cognitive diagnosis because it is based on signed hypergraphs. I find this claim unconvincing. **The essence of the problem remains unchanged: as described in Section 2, the use of signed hypergraph structures to predict student-question responses still reduces to a binary prediction task. The optimization objective is not redefined, nor is a new task proposed** (e.g., transitioning from static classification to temporal forecasting). In fact, the so-called "student performance prediction" task in this work appears to be a degenerated form of cognitive diagnosis. Cognitive diagnosis aims to uncover interpretable student abilities grounded in educational psychology to achieve reliable prediction, whereas student performance prediction here is more of a black-box binary classification with neural networks, lacking interpretability.
> >
> > * **The paper fails to compare against state-of-the-art methods in student ability modeling, particularly those that involve hypergraphs or signed modeling. This omission is problematic and undermines the paper’s empirical rigor.**
> >
> > * Regarding the use of LLMs: while the authors emphasize that this work is an application of machine learning and that LLMs are merely tools, a substantial portion of the claimed contributions in the paper revolves around LLMs (e.g., the third listed contribution, which highlights the novelty of the proposed framework). The model is even named “EduLLM,” which seems inappropriate. **The authors have not made targeted adaptations or innovations that leverage the unique potential of LLMs within the specific educational scenario studied, and thus the technical novelty remains unconvincing.**
> >
> > * **The writing lacks clarity, and several sections suffer from redundancy and poor organization.** For example, the stated motivation — “these methods often fall short in capturing the intricate interactions between students and learning content, as well as the subtle semantics of these interactions” — is questionable. These aspects have been extensively studied in various student performance prediction and cognitive diagnosis works. The paper would benefit from a more thorough and contextually relevant discussion of related work in educational research.
> >
> > Of course, the paper also has its merits, as noted by the other reviewers who gave strong accept recommendations. **However, from the perspective of educational data mining, I believe the current version still has significant issues that need to be addressed before the paper can be considered for acceptance.**
> >
> > ***
> >
> > [1] Shen J, Qian H, Liu S, et al. Capturing Homogeneous Influence among Students: Hypergraph Cognitive Diagnosis for Intelligent Education Systems[C]//Proceedings of the 30th ACM SIGKDD Conference on Knowledge Discovery and Data Mining. 2024: 2628-2639.
> >
> > [2] Shao P, Yang Y, Gao C, et al. Exploring Heterogeneity and Uncertainty for Graph-based Cognitive Diagnosis Models in Intelligent Education[J]. arXiv preprint arXiv:2403.05559, 2024.
> >
> > [3] Qian H, Liu S, Li M, et al. ORCDF: An Oversmoothing-Resistant Cognitive Diagnosis Framework for Student Learning in Online Education Systems[C]//Proceedings of the 30th ACM SIGKDD Conference on Knowledge Discovery and Data Mining. 2024: 2455-2466.

---

> > > ### Author Response · Authors · 2025-04-07
> > >
> > > >- **[On the Novelty of Using Signed Hypergraphs]**: We thank the reviewer for pointing out these related works [1–3] that explore hypergraph-based approaches for student modeling. However, it is important to clarify that these works focus on hypergraphs **without incorporating signed information**, that is, they **do not model signed hyperedges** that explicitly distinguish between correct and incorrect student responses. In contrast, our work introduces **a new problem formulation based on signed hypergraphs for student performance prediction**, where polarity (correctness) is embedded directly into the structural representation. We would also like to highlight that our proposed FraS-HNN outperforms models that only use unsigned hypergraph structures, as demonstrated in the supplementary experiments (**see our first-round rebuttal responses, which we sincerely appreciate your engagement with**). This empirical advantage further supports the relevance and utility of our signed formulation for this task.
> > >
> > > >- **[Distinction from Cognitive Diagnosis Task]**: We agree that both student performance prediction and cognitive diagnosis aim to predict student outcomes, but we respectfully emphasize that the problem formulations are distinct. CD is typically grounded in interpretable latent traits, Q-matrices, or concept mappings, whereas **our formulation makes no such assumption**. Moreover, our used benchmark datasets for performance prediction **do not contain concept labels or Q-matrices**, making them incompatible with traditional CD frameworks. Our task focuses on question-level prediction by modeling correctness-aware interactions through signed hypergraphs, without requiring additional latent or concept-level supervision. While both tasks may produce binary outcomes, **our goal is not to replace cognitive diagnosis but to offer a structure-based, polarity-sensitive alternative with a different modeling philosophy**.
> > >
> > > >- **[On Comparisons with Other Related Works in Student Ability Modeling]**: We appreciate the recommendation to include comparisons with recent works such as ORCDF [3] or HCD [1]. As we clarified in our previous response, many of these works are based on fundamentally different problem settings (e.g., concept-aware diagnosis or sequential modeling), and thus not directly aligned with our structure-based, question-level prediction task. That said, **we agree that acknowledging and discussing these works in more detail would enhance the completeness of the related work section, and we will incorporate a contextualized comparison in a future revision**. We also believe our additional comparisons with classic hypergraph neural network baselines (i.e., HGNN, HyperGCN, AllDeepSets, AllSetTransformer, ED-HNN, and SheafHyperGNN) already validate the effectiveness of FraS-HNN under fair structural settings.
> > >
> > > >- **[On the Role of LLMs and the Naming of EduLLM]**:  We appreciate the reviewer’s perspective and understand the concern regarding the role of LLMs in our framework. As clarified previously, our primary technical contribution lies in the structural design of FraS-HNN, while LLMs are used as 'off-the-shelf' tools to extract semantic embeddings for MCQs. The name "EduLLM" was chosen not to suggest innovation in LLM modeling itself, but rather to reflect the integration of semantic representations from LLMs with signed hypergraph-based structural modeling for educational applications. We hope that this concise naming does not cause confusion, but instead helps motivate future research into more tightly coupled or co-designed methods that combine LLMs and hypergraph learning, potentially leading to new frameworks for diverse application scenarios.
> > >
> > > >- **[On Writing Clarity]**:  We appreciate the feedback and agree that the motivation could be made more context-aware and tightly linked to existing educational research. While our intention was to provide a structural perspective on modeling correctness-aware interactions, we will revise the introduction to more carefully position our work relative to both cognitive diagnosis and student modeling literature. We also acknowledge redundancy in some sections and will revise the manuscript to improve clarity and organization in the final version.
> > >
> > > > Finally, we sincerely thank the reviewer for the prompt, detailed, and constructive follow-up discussion. From our perspective, **your comments, particularly from the angle of educational data mining and/or student modeling, complement the feedback from other reviewers**, who primarily focused on the hypergraph representation learning aspects of our work. This diversity of perspectives has helped us better position, clarify, and strengthen the contributions of our paper (**which will be reflected in the updated version**), and we are truly grateful for the thoughtful engagement.
> > > If there are any remaining questions, we are happy to engage in further discussion and provide additional explanations where needed.

---

### Decision · Program_Chairs · 2025-05-01

**Decision:**

Accept (poster)

**Comment:**

This paper presents EduLLM, a framework that combines large language models (LLMs) with hypergraph learning to improve student performance prediction. Most reviewers found the paper is well organized and recognized the technical contributions of this work. Meanwhile, reviewers raised some concerns about motivation, technical details, baselines, paper writing, etc. Most of these concerns were successfully addressed by the author's rebuttal.